# A MAC2-positive progenitor-like microglial population is resistant to CSF1R inhibition in adult mouse brain

Lihong Zhan[1†], Li Fan[2†], Lay Kodama[1,2,3,4], Peter Dongmin Sohn[1], Man Ying Wong[2], Gergey Alzaem Mousa[2], Yungui Zhou[1], Yaqiao Li[1], Li Gan[1,2,3*]

[1]Gladstone Institute of Neurological Diseases, San Francisco, United States; [2]Helen and Robert Appel Alzheimer's Disease Institute, Brain and Mind Research Institute, Weill Cornell Medicine, New York, United States; [3]Neuroscience Graduate Program, University of California, San Francisco, San Francisco, United States; [4]Medical Scientist Training Program, University of California at San Francisco, San Francisco, United States

**Abstract** Microglia are the resident myeloid cells in the central nervous system (CNS). The majority of microglia rely on CSF1R signaling for survival. However, a small subset of microglia in mouse brains can survive without CSF1R signaling and reestablish the microglial homeostatic population after CSF1R signaling returns. Using single-cell transcriptomic analysis, we characterized the heterogeneous microglial populations under CSF1R inhibition, including microglia with reduced homeostatic markers and elevated markers of inflammatory chemokines and proliferation. Importantly, MAC2/*Lgals3* was upregulated under CSF1R inhibition, and shared striking similarities with microglial progenitors in the yolk sac and immature microglia in early embryos. Lineage-tracing studies revealed that these MAC2+ cells were of microglial origin. MAC2+ microglia were also present in non-treated adult mouse brains and exhibited immature transcriptomic signatures indistinguishable from those that survived CSF1R inhibition, supporting the notion that MAC2+ progenitor-like cells are present among adult microglia.

**\*For correspondence:**
lig2033@med.cornell.edu

[†]These authors contributed equally to this work

## Introduction

Microglia are the primary innate immune cells in the CNS, capable of mounting inflammatory responses and phagocytosis. They can be distinguished from other CNS cell types by their distinctive ramified morphology and expression of common myeloid markers including CD11b and ionized calcium-binding adaptor molecule 1 (IBA1). In addition to immune functions, microglia carry out a multitude of neurotrophic functions during CNS development and homeostasis (*Kierdorf and Prinz, 2017*). Microglia also play critical pathological roles in a wide spectrum of neurodegenerative conditions, including Alzheimer's disease, Parkinson's disease, Huntington's disease, and Amyotrophic lateral sclerosis (*Hickman et al., 2018*). A number of disease genes were found to be highly expressed in microglia (*Hansen et al., 2018*), such as *Trem2* (*Abduljaleel et al., 2014*; *Jonsson et al., 2013*) and *Grn* (*Baker et al., 2006*), highlighting the importance of microglia in neurodegenerative diseases.

Unlike other CNS glial cells, microglia originate from the embryonic mesoderm and follow a convoluted developmental journey (*Rezaie and Male, 2002*). It starts with the emergence of c-kit+ erythromyeloid progenitors in the yolk sac, known as primitive hematopoiesis, which then influx into the developing parenchyma via circulation (*Ginhoux et al., 2010*) in an IRF-8, PU.1-dependent manner (*Kierdorf et al., 2013*). Seeded microglial progenitors persist in the CNS and continue to expand and mature until adulthood (*Matcovitch-Natan et al., 2016*). In general, developing

microglia can be distinguished by well-defined developmental intervals from the yolk sac to the adult, and transcriptional programs associated with each of these stages have been meticulously mapped (*Matcovitch-Natan et al., 2016*; *Hammond et al., 2019*). In particular, homeostatic maturation in microglia requires the transcription factor MAFB (*Matcovitch-Natan et al., 2016*) as well as TGF-beta signaling (*Butovsky et al., 2014*; *Zöller et al., 2018*), and can be distinguished by homeostatic markers such as *Tmem119* (*Bennett et al., 2016*) and *P2ry12* (*Haynes et al., 2006*). Interestingly, we recently discovered that adult newborn microglia follow a similar maturation path (*Zhan et al., 2019*), suggesting that the developmental plasticity of microglia in the adult brain might be an underlying feature of microglial homeostasis (*Santambrogio et al., 2001*). Unlike other tissue myeloid populations such as monocytes and macrophages, the resident microglial pool receives no significant replenishment from circulation and is internally maintained by self-renewal (*Mildner et al., 2007*; *Ajami et al., 2007*), even under conditions of acute ablation (*Bruttger et al., 2015*; *Huang et al., 2018*; *Zhan et al., 2019*). It is thus not surprising that microglia have an extremely long half-life (*Lawson et al., 1992*), most recently estimated to be 7.5–15 months in the murine CNS (*Tay et al., 2017*; *Füger et al., 2017*; *Zhan et al., 2019*). In contrast, other myeloid populations such as classical monocytes have a half-life of less than 24 hr (*van Furth and Cohn, 1968*; *Yona et al., 2013*), and require constant replenishment from a CX3CR1- population in the bone marrow (*Fogg et al., 2006*).

CSF1R signaling is critical for microglial survival and maintenance. Loss-of-function mutations in either of its two natural ligands, CSF1, and IL-34, results in a significant reduction in microglia number (*Wegiel et al., 1998*; *Greter et al., 2012*). Null mutations in *Csf1r* remove 99.7% microglia, while a few morphologically-distinctive microglia near the hippocampus and piriform cortex remain intact (*Erblich et al., 2011*). In addition, the CSF1R inhibitor PLX5622 (PLX) has been widely used as a research tool to acutely deplete microglia. While depletion efficiency varies, complete microglial ablation has never been reported (*Acharya et al., 2016*; *Rice et al., 2017*; *Huang et al., 2018*; *Zhan et al., 2019*). These studies suggest that the adult microglial pool includes a population that does not require CSF1R signaling for survival. Remarkably, the microglial pool can be rapidly regenerated after termination of PLX administration. While an earlier study proposed a hidden NESTIN+ progenitor pool responsible for repopulation (*Elmore et al., 2014*), we and others found that the remaining microglia were solely responsible for microglial repopulation (*Huang et al., 2018*; *Zhan et al., 2019*), consistent with the notion that a subpopulation of microglia exhibit progenitor-like features. Our current study applied single-cell RNA-sequencing (scRNA-seq) to examine this resilient microglial population after acute CSF1R inhibition with PLX. We found that among the PLX-resistant microglia, there was a subpopulation of cells with high-expression of *Mac2* that displayed immature microglial gene signatures. Interestingly, this subpopulation also existed in the homeostatic microglial pool. Together, our data uncovered a progenitor-like microglial state that likely contributes to the basal proliferation of microglia in the brain.

## Results

### Single-cell RNA-seq profiling of microglia under CSF1R inhibition and early repopulation

To examine the transcriptome profiles of the CSF1R inhibitor-resistant microglial population, we performed scRNA-seq on the remaining microglia from six C57BL/6J mice (two brains/sample) that were treated with PLX diet (D0) (*Figure 1a*). To investigate early stage adult newborn microglia, we included an additional four mice (two brains/sample) that were switched to a control diet for 2 days after PLX treatment (D2) (*Figure 1a*). Microglia from three age-matched non-treated mice were used as controls (Ctrl) (*Figure 1a*). Similar to our previous results (*Zhan et al., 2019*), oral dosing of 1200 mg/kg PLX in C57BL/6J mice for two weeks resulted in about 90% removal of CD11b+ myeloid cells in the CNS (*Figure 1—figure supplement 1*). Single-cell suspensions were stained with CD11b antibody for myeloid population purification via fluorescence activated cell sorting (FACS) (*Figure 1—figure supplement 1*), followed by the 10x Genomics single-cell RNA-seq platform (*Figure 1b*).

After filtering the data using defined quality-control metrics (*Figure 1—figure supplement 2*), a total of 28,649 cells were obtained from 13 mice (3 Ctrl mice, 6 D0 mice, 4 D2 mice), at a sequencing depth of about 46,000 reads/cell (*Figure 1—figure supplement 2*). The sequencing quality was

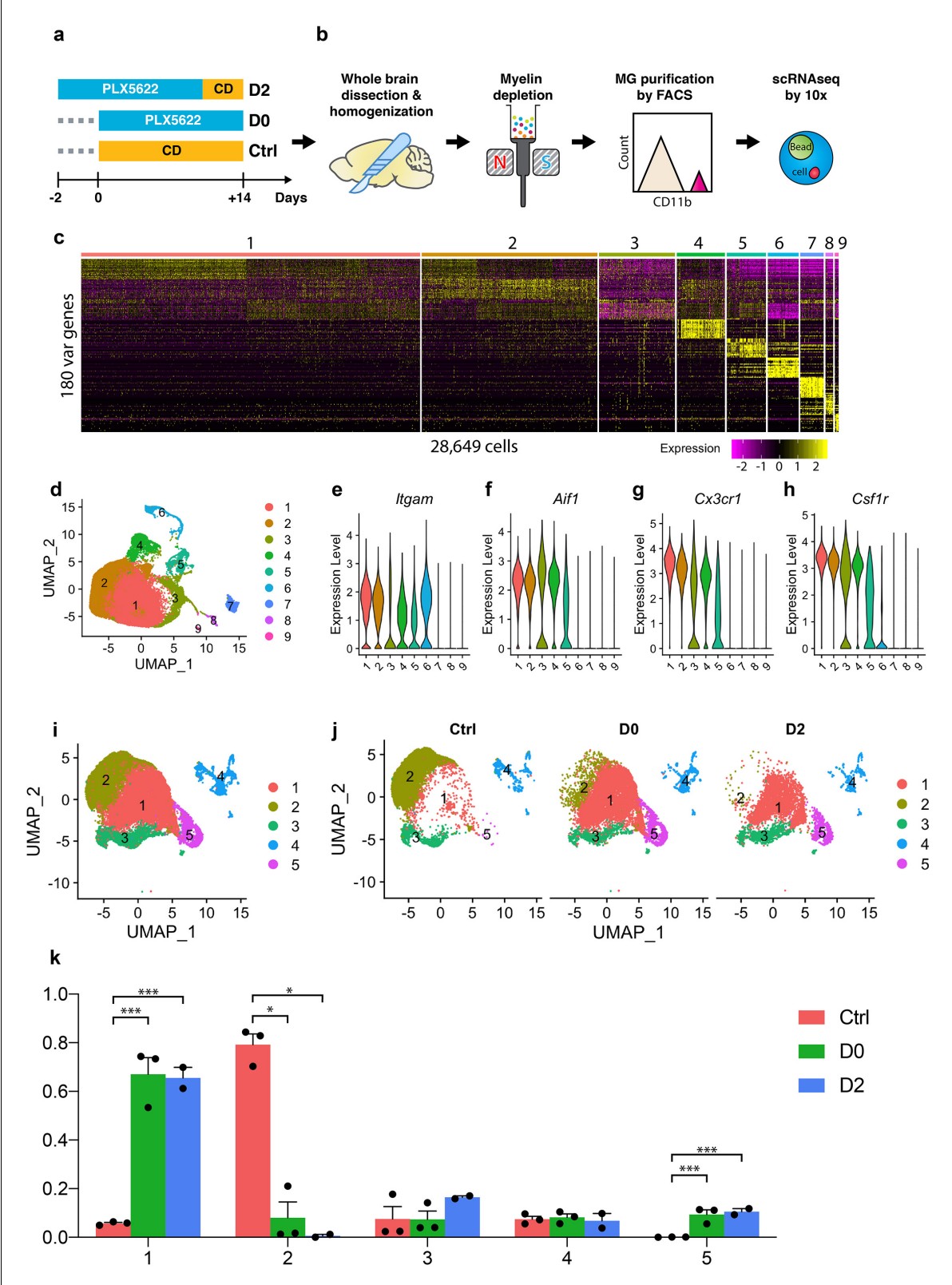

**Figure 1.** Single-cell RNA-seq profiling of microglia under CSF1R inhibition and early repopulation. (a) Experimental design for microglia depletion and repopulation. Mice were placed on PLX diet (1200 mg/kg) for 14 days to deplete microglia (D0). The early stage microglial repopulation (D2) group was switched to control diet (CD) for 2 days. Microglia from each mouse were collected on the same day. A total of 15 female C57BL/6J mice (5 Mo) were used: Ctrl (n = 3); D0 (n = 6, two brains pooled together for FACS); D2 (n = 4, two brains pooled together for FACS); (b) Workflow of adult microglia

*Figure 1 continued on next page*

*Figure 1 continued*

isolation procedures for scRNA-seq capture. Detailed description in methods. (**c**) Heatmap showing the top 180 variable genes detected from 28,649 cells after initial data filtering. (**d**) UMAP plot showing nine distinctive clusters identified from the scRNA-seq data. (**e–h**) Violin plot showing expression of *Itgam, Aif1, Cx3cr1,* and *Csf1r in* all clusters. Clusters 6, 7, 8, and 9 were removed from downstream data analysis. (**i**) Reclustered UMAP after removal of clusters 6–9 showing five distinctive clusters. (**j**) UMAP split by experimental conditions. (**k**) Ratio of cells from three treatment groups distributed in each cluster. Ratio of cells was calculated by normalizing to the total number of cells captured in each sample (n = 3 for Ctrl and D0, two for D2). Data shown as mean ± SEM. p-values were calculated using the negative binomial generalized linear model from EdgeR. *p≤0.05; ***p≤0.001.

The online version of this article includes the following source data and figure supplement(s) for figure 1:

**Source data 1.** Cell distribution in each cluster.
**Figure supplement 1.** FACS strategy for single-cell microglia isolation.
**Figure supplement 2.** scRNA-seq data quality-control metrics.
**Figure supplement 3.** Expression of selected marker genes in clusters 6, 7, 8, and 9.
**Figure supplement 4.** Justification of the removal of sample D2-3.

comparable between samples, as evidenced by total number of UMI counts, genes, and percentage of mitochondrial genes per cell (*Figure 1—figure supplement 2*). Following initial principle component analyses (*Figure 1—figure supplement 2*) and further dimensionality reduction using Uniform Manifold Approximation and Projection (UMAP) (*Becht et al., 2019*), we identified nine distinctive cell clusters (*Figure 1c,d*). Clusters 1–5 expressed *Cd11b, Cx3cr1* and *Csf1r*, three myeloid markers, and *Iba1*, a microglia and macrophage-specific calcium-binding protein, while Clusters 6–9 did not (*Figure 1e–h*). Instead, Clusters 6–9 expressed neutrophil, lymphoid, endothelial and astroglial cell markers, respectively (*Figure 1—figure supplement 3*). We re-performed clustering after the removal of contaminating cells and identified 5 clusters with 25,954 cells (*Figure 1i*). Comparing microglial cells from Ctrl, D0, and D2 samples, we found distinct distributions; those from Ctrl brains were enriched in Cluster-2, while those from D0 and D2 brains were enriched in Clusters 1 and 5 (*Figure 1j,k*, *Figure 1—source data 1*).

## Cluster identification reveals heterogenous microglial populations under CSF1R inhibition

Focusing on microglial Clusters 1–5, we next performed differential expression analysis to identify marker genes for each cluster (*Figure 2a*, *Figure 2—source data 1*). Each cluster could be characterized by a set of high-expressing markers (*Figure 2a*, *Figure 2—figure supplement 1*).

Cluster-1 exhibited high-levels of inflammatory chemokines, including NF-κB target genes such as *Ccl3, Ccl4,* and *Egr1* (*Figure 2a,b*, *Figure 2—figure supplement 1*). Cluster-2 resembled homeostatic microglia, expressing high levels of homoeostatic genes including *P2ry12, Tmem119,* and *Trem2*, while other clusters exhibited reduced expression of homeostatic microglial signatures. Cluster-3, which was enriched in D2 samples, expressed higher levels of genes related to translation and cytoskeleton networks, including Leucyl-tRNA Synthetase 2 (*Lars2*), Dystonin (*Dst*), Microtubule Actin Crosslinking Factor 1(*Macf1*), and ribosomal genes (*Rps26, Rps8*) (*Figure 2c*, *Figure 2—figure supplement 1*). Cluster-5 cells, enriched in both D0 and D2 brains, had upregulation of genes involved in mitosis and proliferation, including Marker of Proliferation Ki-67 (*Mki67*) and DNA Topoisomerase 2a (*Top2a*) (*Figure 2e*, *Figure 2—figure supplement 1*). Other proliferation markers upregulated in this cluster included Stathmin 1 (*Stmn1*) and Ubiquitin Conjugating Enzyme E2 C (*Ube2c*), a member of the anaphase-promoting complex/cyclosome and regulates cell cycle progression (*Figure 2—figure supplement 1*). Consistent with this population representing proliferative microglia, the relative frequency of cells in Cluster-5 was markedly increased from 0.12% in Ctrl to approximately 10% in D0 and D2 groups (*Figure 1f*). Cluster-4 cells expressed MHC genes such as *H2-Ab1, H2-Eb1,* and *Cd74*, a cell surface receptor for the cytokine macrophage migration inhibitory factor (Mif). The expression of these marker genes was also elevated in microglia from D0 and D2 brains (*Figure 2d*). Consistent with this, Cd74-positive cells were largely part of the Cluster-4 subpopulation (*Figure 2f*), and immunostaining for CD74 showed over-representation of CD74-positive microglia in D0 brains, confirming that this cell population is relatively resistant to CSF1R inhibitors (*Figure 2g*).

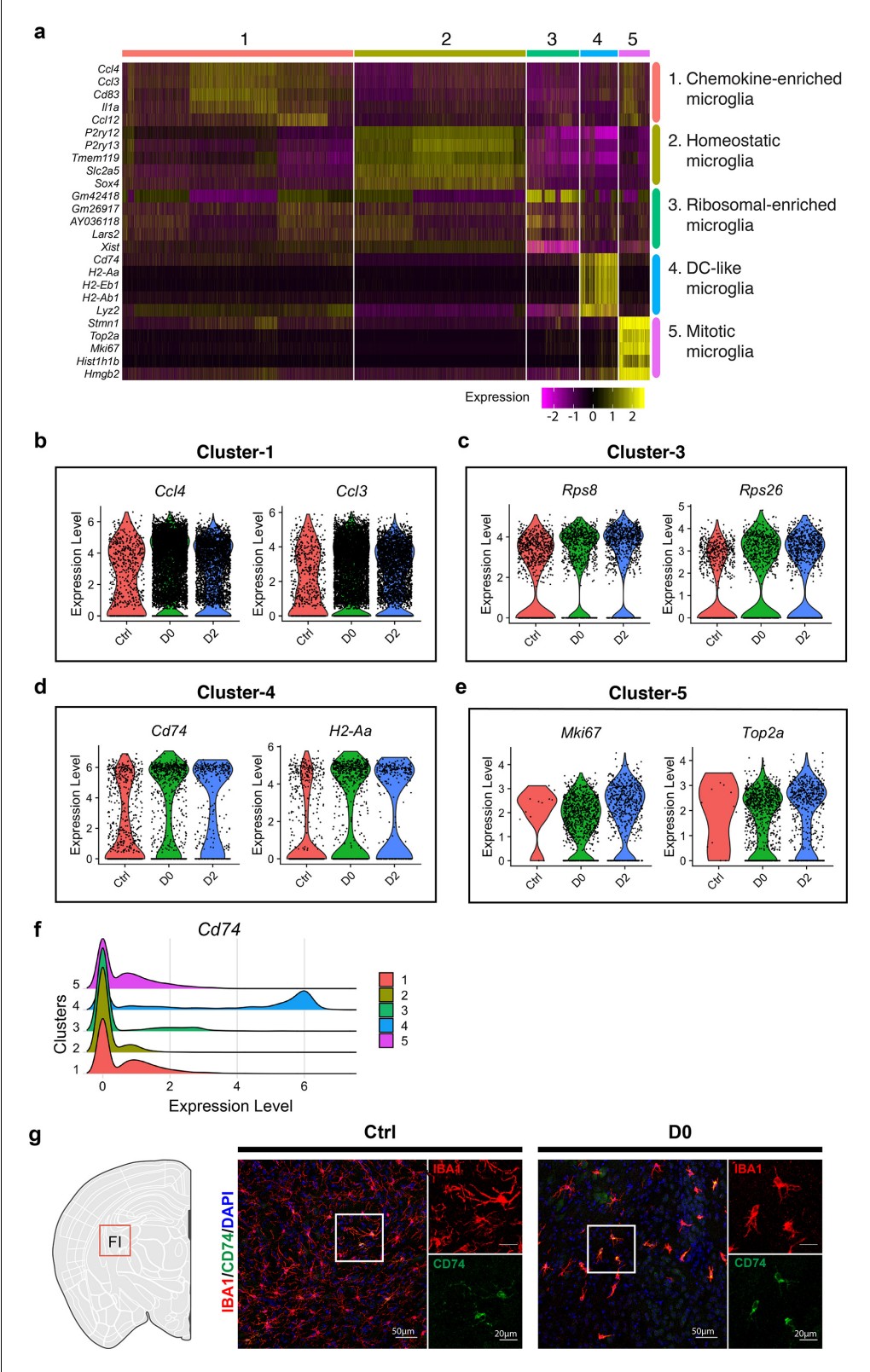

**Figure 2.** Cluster identity reveals heterogenous microglial population. (**a**) Heatmap showing top-5 marker genes identified from each cluster. Complete list of marker genes can be found in *Figure 2—source data 1*. Annotation for each cluster is shown on the right. (**b–e**) Violin plot showing expression of selected marker genes in Cluster-1 (**b**), Cluster-3 (**c**), Cluster-4 (**d**) and Cluster-5 (**e**) in naïve C57BL/6J mice (Ctrl), PLX-treated mice (D0, 2 weeks of PLX

*Figure 2 continued on next page*

*Figure 2 continued*

diet) and repopulating (D2, 2 days after the end of treatment) brains. (f) Ridge plot showing expression of *Cd74* in clusters 1–5. (g) Representative confocal images showing immunofluorescence staining of CD74 (green) and IBA1(red) in Ctrl and D0 brains. Images were taken from the fimbria (FI).

The online version of this article includes the following source data and figure supplement(s) for figure 2:

**Source data 1.** DEGs for each cluster vs all other clusters.
**Figure supplement 1.** Expression of Top DEGs in microglial clusters 1–5.

## Microglial homeostatic signatures are reduced in remaining microglia under PLX treatment and early stage repopulation

Compared to cells in control mice, cells in D0 and D2 samples had significant downregulation of microglial homeostatic genes, such as *Tmem119* (*Figure 3a*), *P2ry12* (*Figure 3b*), *P2ry13* (*Figure 3c*), *Seplg* (*Figure 3d*), *Cx3cr1* (*Figure 3e*) and *Csf1r* (*Figure 3f*). Furthermore, this downregulation of microglial homeostatic genes was observed within every cluster (*Figure 3—figure supplement 1*). To validate this observation, we immunostained D0 and Ctrl brains for P2RY12 and TMEM119 (*Figure 3g*). The fluorescence intensity of P2RY12 and TMEM119 per IBA1+ cell was significantly decreased in D0 microglia compared to Ctrl microglia (*Figure 3h,i*), consistent with the transcriptomic findings.

## Mac2 expression in the CSF1R inhibition-resistant microglial population

We next performed differential expression analysis of D0 vs. Ctrl and identified *Il-1β*, *lyz2*, *Cd52*, *Cd74*, and *Mac2*, also known as Galectin 3 (*Lgals3*), as some of most highly upregulated genes in D0 (*Figure 4a*, *Figure 4—source data 1*). Since MAC2 was shown to be a ligand for Trem2 signaling (*Boza-Serrano et al., 2019*), a pathway required for microglial homeostatic maintenance (*Butovsky et al., 2014*; *Zöller et al., 2018*), we then focused on further investigating the *Mac2+* subpopulation. Cells with expression of *Lgals3* were distributed in Cluster-1, which was highly enriched in D0 and D2, and Cluster-4 (*Figure 4b*). Although there is no significant difference in abundance within Cluster-4 cells across conditions (*Figure 1k*), there are substantially more *Lgals3+* cells relative to the total microglia population at D0 and D2 (*Figure 4c*). Immunofluorescence staining of D0 brain tissues revealed a subset of IBA1+ microglia expressing MAC2 in both Ctrl and D0 groups, with D0 microglia having increased MAC2 intensity and a more ramified morphology (*Figure 4d*). After 2 weeks of PLX treatment, while 88% of IBA1+ cells were depleted (*Figure 4e*), the number of IBA1+MAC2+ cells was modestly increased (D0) in the hippocampus (*Figure 4f*). Notably, MAC2+ microglia accounted for approximately 0.68% ± 0.16 (SEM) of all IBA1+ microglia in adult hippocampus at baseline while accounting for 15.46% ± 4.55 (SEM) of all remaining microglia under CSF1R inhibition (*Figure 4g*). The dramatic increase in the percentage of MAC2+ microglia induced by CSF1R inhibition supports the notion that this subpopulation of microglia can survive without CSF1R signaling.

## MAC2+ microglia are not derived from peripheral monocytes and are highly proliferative

Since MAC2 is a galactoside-binding protein expressed in many myeloid cells including monocytes and macrophages (*Ho and Springer, 1982*), one possibility is that IBA1+MAC2+ cells are over-represented under CSF1R inhibition due to monocytic replenishment from circulation. To test this hypothesis, we performed lineage tracing using the myeloid specific *CX3CR1-CreERT2* driver (*Parkhurst et al., 2013*) with an inducible DsRed reporter (*Luche et al., 2007*) (*CX3CR1-CreERT2/Rosa26-stop-DsRed*). One month after the initial tamoxifen injection, 98% of labeled monocytes in circulation are replaced by unlabeled newborn cells, whereas microglia in the brain remain RFP+ due to their extreme longevity (*Parkhurst et al., 2013*). Using a similar labeling strategy, we examined the lineage of the parenchymal MAC2+ cells (*Figure 5a*). We reasoned that if the MAC2+ microglia were derived from circulating monocytes/macrophages that lack RFP expression, there would be fewer MAC2+ microglia that express RFP after repeated PLX treatment. Instead, RFP labeling efficiency in the MAC2+ population did not differ after either single (PLX[1X]) or tandem PLX treatment (PLX[2X]) separated by a repopulation period (*Figure 5b–d*), indicating that the MAC2+ cells are not

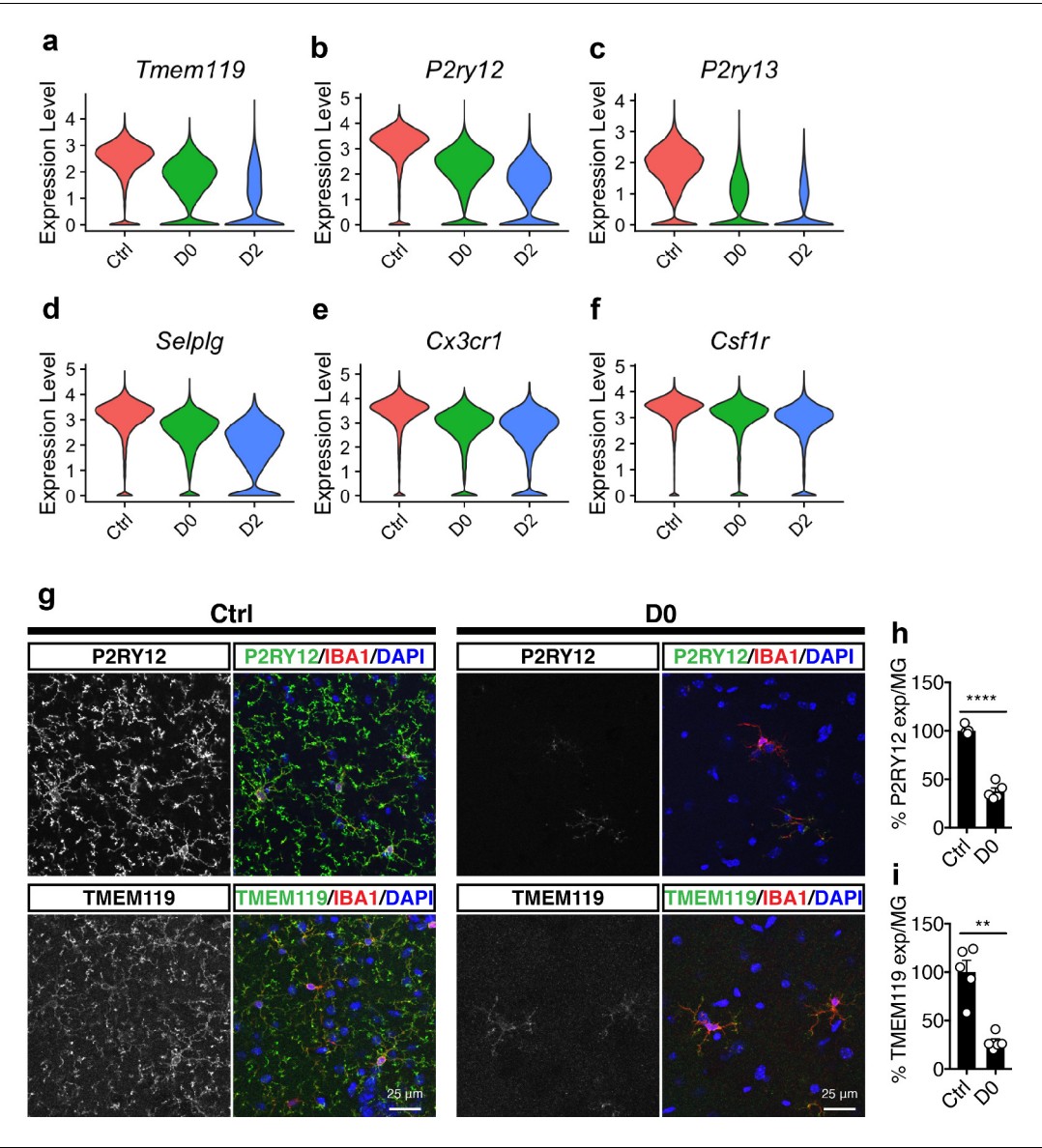

**Figure 3.** Microglial homeostatic signatures are down-regulated under CSF1R inhibition. (**a–f**) Violin plots showing expression level (log-transformed total UMI) of selected homeostatic genes: (**a**) *Tmem119*; (**b**) *P2ry12*; (**c**) *P2ry13*; (**d**) *Selplg*; (**e**) *Cx3cr1*; (**f**) *Csf1r*. (**g**) Representative confocal images showing P2RY12 and TMEM119 expression in naive mice (Ctrl) and PLX-treated mice (D0, 2 weeks of PLX diet). IBA1 was used as marker for microglia. Images were taken from the hippocampal region. (**h**) Quantification of relative P2RY12 expression level per microglial cell. Number of C57BL/6J mice (3–5 Mo) used: Ctrl (n = 5), D0 (n = 5). Unpaired t-test with Welch's correction was used. (**i**) Quantification of relative TMEM119 expression level per microglial cell. Number of C57BL/6J mice (3–5 Mo) used: Ctrl (n = 5), D0 (n = 5). Unpaired t-test was used. p-value summary is shown as ns (p>0.05); * (p≤0.05); ** (p≤0.01); *** (p≤0.001); **** (p≤0.0001).

The online version of this article includes the following figure supplement(s) for figure 3:

**Figure supplement 1.** Microglial homeostatic signatures are down-regulated across all clusters.

derived from circulating monocytes, but rather resident microglia that are internally maintained in the parenchyma.

We then examined the newborn microglia at day 4 and day 14 after switching mice from a PLX diet to the control diet using 5-Ethynyl-2'-deoxyuridine (EdU) pulse-chase labeling. To maximize the

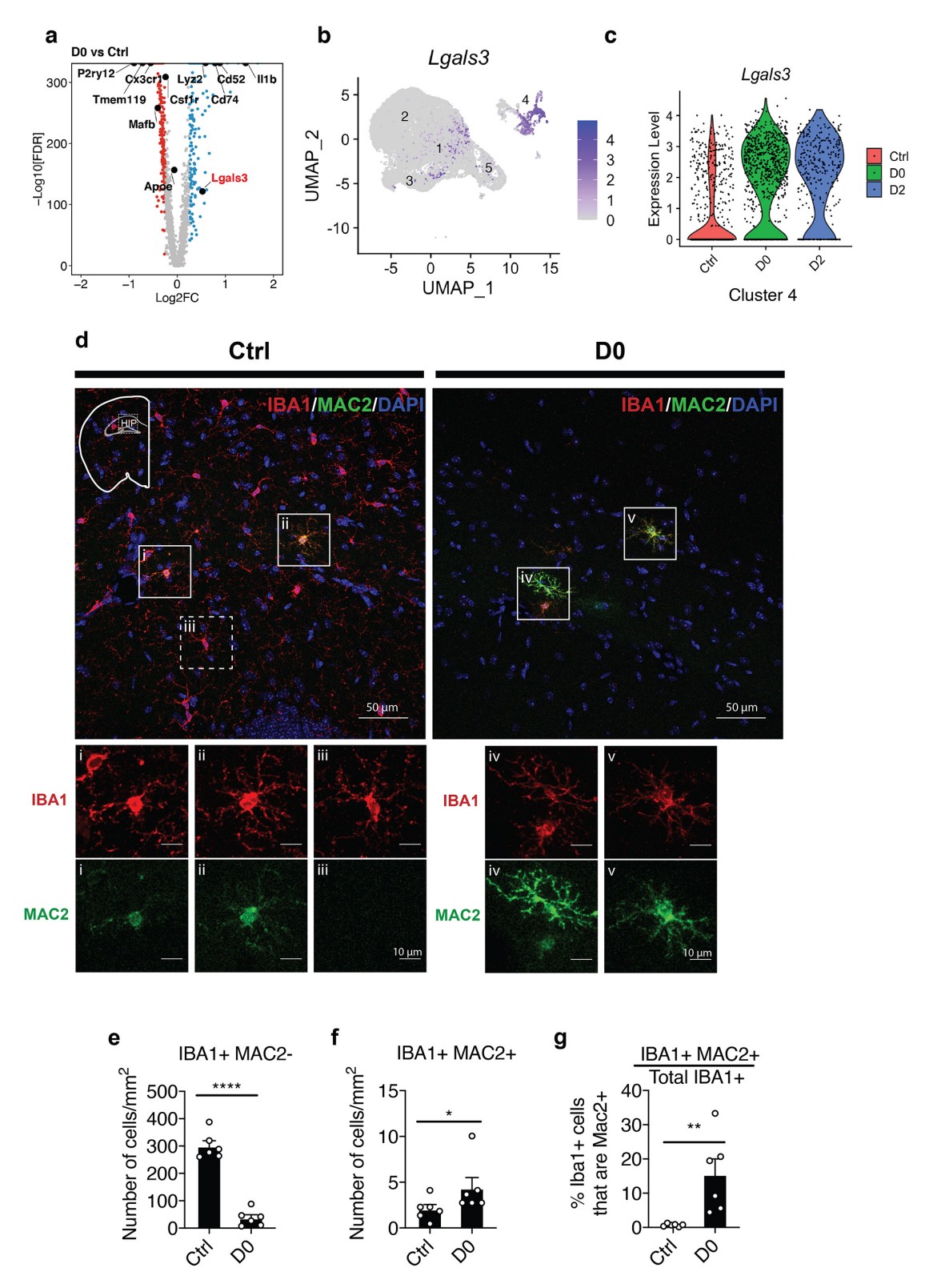

**Figure 4.** Microglia populations resistant to CSF1R inhibition express MAC2 (Galectin-3). (a) Volcano plot showing differentially expressed genes identified by comparison of D0 vs. Ctrl. Genes of interest are in text, with *Lgals3* (encodes for the MAC2 protein) highlighted in red. Upregulated DEGs are colored in red and downregulated DEGs are colored in blue (log fold change >0.25 or<−0.25). Complete list of DEGs can be found in *Figure 2— source data 1*. (b) UMAP plot showing expression distribution of *Lgals3* in all clusters. Log-transformed total UMI was plotted. (c) Violin plot showing

*Figure 4 continued on next page*

Figure 4 continued

expression of *Lgals3* in Cluster-4 in each treatment group. (d) Representative confocal images showing colocalization of IBA1 and MAC2 in native mice (Ctrl) and PLX-treated mice (D0, 2 weeks of PLX diet). Images were collected from the hippocampal region (shown in mini-map). Solid box highlights cell that is IBA1+MAC2+ and dotted box highlights cell that is IBA1+MAC2-. Enlarged images from the boxed area are shown in separate channels (panel i-v). (e) Quantification of IBA1+MAC2- cell numbers. Unpaired t-test was used. (f) Quantification of IBA1+MAC2+ cell numbers. Mann Whitney test was used. (g) Quantification of the percentage of MAC2+ cells among all IBA1+ microglia. Unpaired t-test was used. Number of C57BL/6J mice (2.5–4 Mo) used (panel e-g): Ctrl (n = 5); D0 (n = 6). Quantification in (panel e-g) was performed on images (1131.56 µm x 1938.59 µm) collected at the hippocampus using the VERSA automated slide scanner (Leica, 20x lens).

The online version of this article includes the following source data for figure 4:

**Source data 1.** DEGs for D0 vs Ctrl.

EdU labeling efficiency, we gave two intraperitoneal injections separated by 7 hr on repopulation day 3 (*Figure 5e*). In the ectorhinal cortex, we confirmed microglia repopulation (*Figure 5f–h*) and found that, while a subset of both MAC2+ and MAC2– microglia have incorporated Edu (*Figure 5f, g*), the number of IBA1+MAC2+ cells increased substantially at repopulation day 4 (D4) (*Figure 5i*), accounting for ~5% of total IBA1+ microglia (*Figure 5j*). Roughly 50% of the IBA1+MAC2+ cells were EdU+ at repopulation day 4 (*Figure 5k*). Their highly proliferative property was further supported by the presence of mitotic marker KI67 (*Figure 5—figure supplement 1*). Notably, newborn cells from the IBA1+MAC2+ population failed to retain long-term MAC2+ expression, and the number of IBA1+MAC2+ cells dropped back to homeostatic levels after repopulation day 14 (*Figure 5i*).

## MAC2+ cells display microglial immature signatures

To further characterize the MAC2+ microglial population, we subset the MAC2+ cells from our scRNA-seq dataset by defining MAC2+ cells as having *Lgals3* expression one standard deviation above the average log UMI counts. Overlay of the MAC2+ cells on the UMAP plot showed that they were largely derived from Cluster-4, with some cells distributed in the chemokine-enriched Cluster-1 and ribosomal-enriched Cluster-3 (*Figure 6a*). A total of 1937 cells were manually distinguished and binned together as the MAC2+ cluster (*Figure 6b*), of which 1080 were from Cluster-4, while 653 and 126 were from the chemokine-enriched and ribosomal-enriched clusters, respectively. To investigate the gene signatures in MAC2+ cells, we next performed DEG analysis in comparison to homeostatic Cluster-2 and identified DEGs up- and down-regulated in MAC2+ cells (*Figure 6c*, *Figure 6—source data 1*). Among the DEGs, early microglial development genes such as *Lyz2*, (*Matcovitch-Natan et al., 2016*), was significantly upregulated, while mature microglia signature genes such as *Tmem119*, *Mafb*, *Cx3cr1*, and *Csf1r* (*Matcovitch-Natan et al., 2016*) were downregulated (*Figure 6c*).

To further examine whether the MAC2+ cells resembled immature microglial progenitors during development, we compared the DEGs of MAC2+ cells with the microglial developmental gene sets identified previously (*Matcovitch-Natan et al., 2016*). In keeping with the original terminology, a total of 7 different gene clusters were compared, which covered the entire microglial developmental trajectory, from yolk sac (Cluster YS, day E10.5 – E12.5), early microglia (Cluster E1/E2, day E10.5 – E14), pre-microglia (Cluster P1/P2, day E14 to postnatal day 9), to adulthood (Cluster A1/A2,>1 Mo) (*Matcovitch-Natan et al., 2016*). Remarkably, genes associated with early development were highly enriched among the upregulated DEGs in the MAC2+ cells (*Figure 6d*). Specifically, 21.91% of upregulated genes were those associated with yolk sac (YS) progenitors and 44.72% were those expressed in early embryonic progenitors (E1), while only around 2.84–5.28% were those expressed in adult microglia (*Figure 6d*). The upregulated genes also showed similar overlap with E14.5 microglial signatures described in a previously-published single-cell RNA-seq study of microglia (*Hammond et al., 2019*; *Figure 6—figure supplement 1*). In contrast, among the downregulated DEGs from MAC2+ cells, 24.30% and 26.16% of them overlapped with the A2 and A1 adult gene signatures respectively (*Figure 6e*). As a control for the analysis, we examined cells lacking MAC2 expression (MAC2–), which exhibited signatures in direct contrast to MAC2+ cells. Specifically, upregulated DEGs in MAC2– cells showed significant overlap with adult microglia (26.26% and 25.10% overlapping with A1 and A2) while downregulated DEGs overlapped with early embryonic (45.15% overlapping with E1) and yolk sac progenitors (19.71%) (*Figure 6g*).

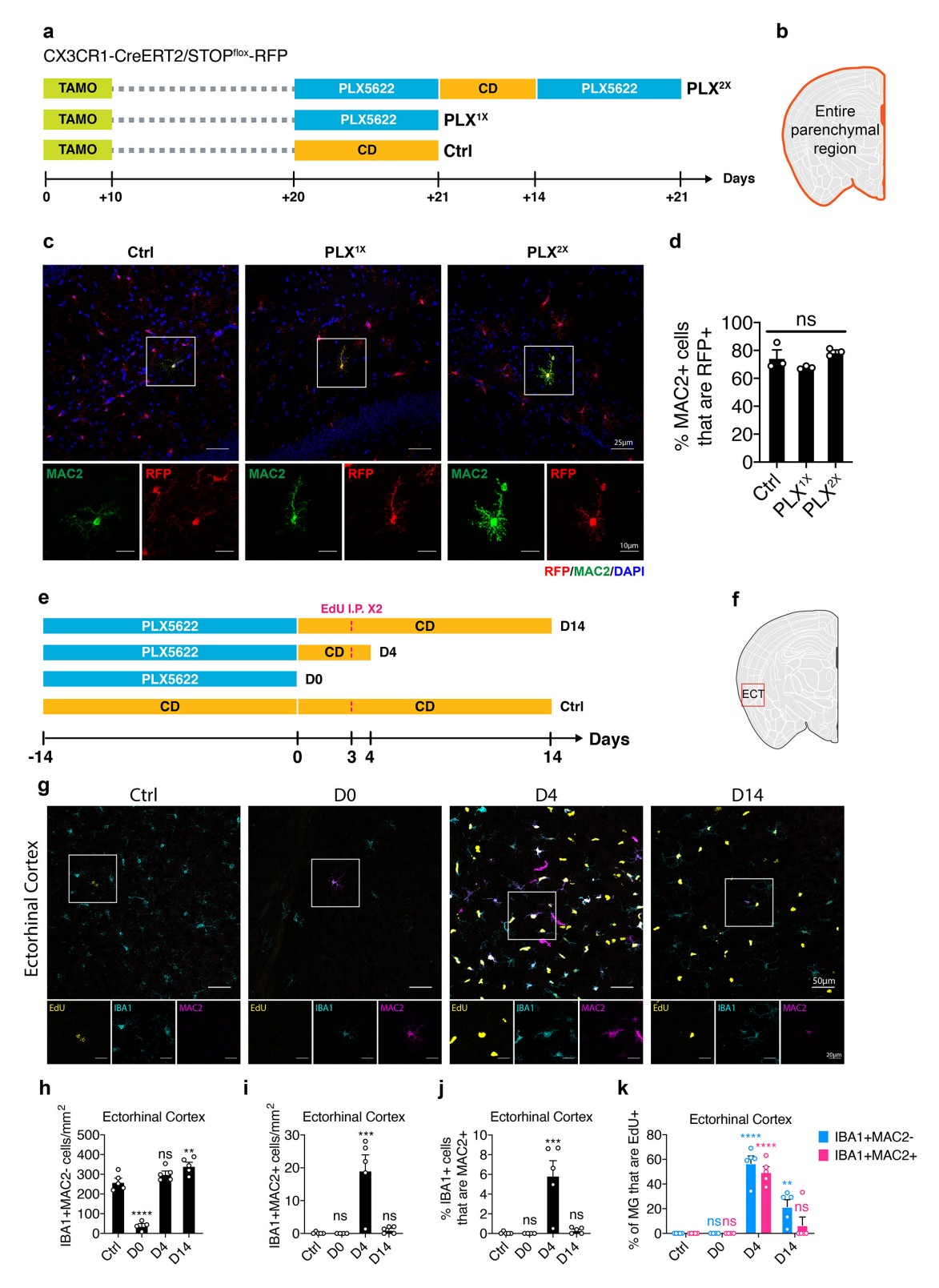

**Figure 5.** Lineage mapping shows MAC2+ microglia are not derived from circulating monocytes. (**a**) Experimental design of the lineage mapping. *Cx3Cr1-CreERT2/Rosa26-stop-DsRed* mice were injected with tamoxifen (10 days) to label microglia with RFP. Mice are either treated with PLX diet for 3 weeks (PLX[1X]) or underwent repopulation for 2 weeks and treated with PLX diet again for another 3 weeks (PLX[2X]). (**b**) Quantification area was performed on the entire parenchymal region. (**c**) Representative confocal images showing colocalization of MAC2 and RFP expression. Boxed area

*Figure 5 continued on next page*

Figure 5 continued

is enlarged and separated by each channel. Images were collected from the hippocampal region. (d) Quantification of the percentage of MAC2+ cell that are RFP+. Number of *CX3CR1-CreERT2/Rosa26-stop-DsRed* mice (7–9 Mo) used: Ctrl (n = 3); PLX[1X] (n = 3); PLX[2X] (n = 3). One-way ANOVA was used. p-value summary is shown as ns (p>0.05); * (p≤0.05); ** (p≤0.01); *** (p≤0.001); **** (p≤0.0001). (e) Experimental design of microglial repopulation timeline and EdU injections. C57BL/6J mice were treated with PLX diet for 2 weeks (D0) and switched to control diet (CD) to start repopulation for 4 days (D4) or 14 days (D14). EdU was injected on repopulation day 3. (f) Brain region used for quantification. Quantification in panel (h–k) was performed on images (1292.23 μm x 1130.7 μm) collected at the Eentorhinal cortex (ECT) using the VERSA automated slide scanner (Leica, 20x lens). (g) Representative confocal images showing immunofluorescence staining of EdU (yellow), IBA1 (cyan), and MAC2 (magenta) in the entorhinal region. Boxed area is shown by separated channels at the bottom. (h) Quantification of IBA1+MAC2- cells in the ECT. (i) Quantification of IBA1+MAC2+ cells in the ECT. (j) Quantification of the percentage of IBA1+ microglia that are MAC2+ in the ECT. (k) Quantification of the percentage of EdU+ labeling in either IBA1+MAC2- cells (blue bar) or IBA1+MAC2+ cells (red bar). Number of C57/BL6 mice (2–3.5 Mo) used: Ctrl (n = 5); D0 (n = 4); D4 (n = 5); D14 (n = 5). Statistical tests used: (1) In panels (h–j), one-way ANOVA with Dunnett's multiple comparisons test was used to compare with Ctrl; (2) In panels (k), two-way ANOVA with Dunnett's multiple comparisons test was used to compare with Ctrl for each cell population. p-value summary is shown as ns (p>0.05); * (p≤0.05); ** (p≤0.01); *** (p≤0.001); **** (p≤0.0001).

The online version of this article includes the following figure supplement(s) for figure 5:

**Figure supplement 1.** IBA1+MAC2+ cells express mitotic marker KI67 during early repopulation.

We next performed Gene Set Enrichment Analysis (GSEA) and upstream regulator predictions using Ingenuity Pathway Analysis (IPA) on the DEGs defining MAC2+ microglia (*Mootha et al., 2003*; *Subramanian et al., 2005*). Genes involved in Tnf signaling via NF-κb and interferon signaling were highly enriched in the DEGs (*Figure 6h*), with these pathways predicted to be activated (*Figure 6i*), whereas TREM2 signaling, a pathway required for microglial homeostatic maintenance (*Butovsky et al., 2014*; *Zöller et al., 2018*), was predicted to be inhibited (*Figure 6i*). TREM2 signaling is mediated via TYROBP, which is the downstream adaptor of CSF1R signaling (*Otero et al., 2009*). Therefore, the downregulation of TREM2 signaling could potentially explain the resistance of MAC2+ cells to CSF1R inhibition.

## MAC2+ progenitor-like cells are present among homeostatic microglia

MAC2+ cells also existed in non-treated Ctrl brains (*Figure 4d*). We therefore asked whether these MAC2+ cells in non-treated Ctrl brains also have progenitor-like features. 3.0% (300/10103) of cells in Ctrl brains were MAC2+, while this percentage increased in treated conditions to 9.8% (1004/10209) and to 11.2% (633/5642) in D0 and D2 brains, respectively (*Figure 7a*). MAC2+ cells from Ctrl mice exhibited similar upregulation of developmental genes and downregulation of homeostatic markers compared to Cluster-2 cells (*Figure 7b*, *Figure 7—source data 1*). Specifically, almost all upregulated DEGs in MAC2+ cells in Ctrl brains were also upregulated DEGs in all MAC2+ cells (1026 genes vs. 151 genes) (*Figure 7c*). Similarly, only 21 genes out of 1664 downregulated DEGs of MAC2+ cells in Ctrl brains were not found in the downregulated DEGs of all MAC2+ cells (*Figure 7d*). These results suggest that MAC2+ cells in the presence and absence of CSF1R inhibition are transcriptomically indistinguishable. Indeed, MAC2+ cells from the control brain exhibited similar microglial immature signatures with elevated expression of embryonic microglial markers (*Figure 7e*), and reduced expression of mature microglial markers (*Figure 7f*), further confirming the presence of MAC2+ progenitor-like microglia in the adult mouse brain.

## Discussion

In the current study, we employed scRNA-seq to characterize the CNS myeloid population under acute CSF1R inhibition and identified a resistant microglia population that expresses MAC2 antigen. In the hippocampus, the MAC2+ microglial sub-population represented 0.68% of total microglia and did not require CSF1R signaling for survival. While the MAC2+ microglia shared transcriptomic features similar to those of circulating monocytes, lineage tracing revealed that the MAC2+ microglial population was self-sustained with no replenishment from the periphery. Remarkably, the MAC2+ population shared striking similarities with immature microglial progenitors during development. Finally, MAC2+ microglia appeared to be highly proliferative in the entorhinal region during adult microglial repopulation after acute ablation. Altogether, our data identified a heterogeneous progenitor-like microglial population that is resistant to CSF1R inhibition in adult mouse brain.

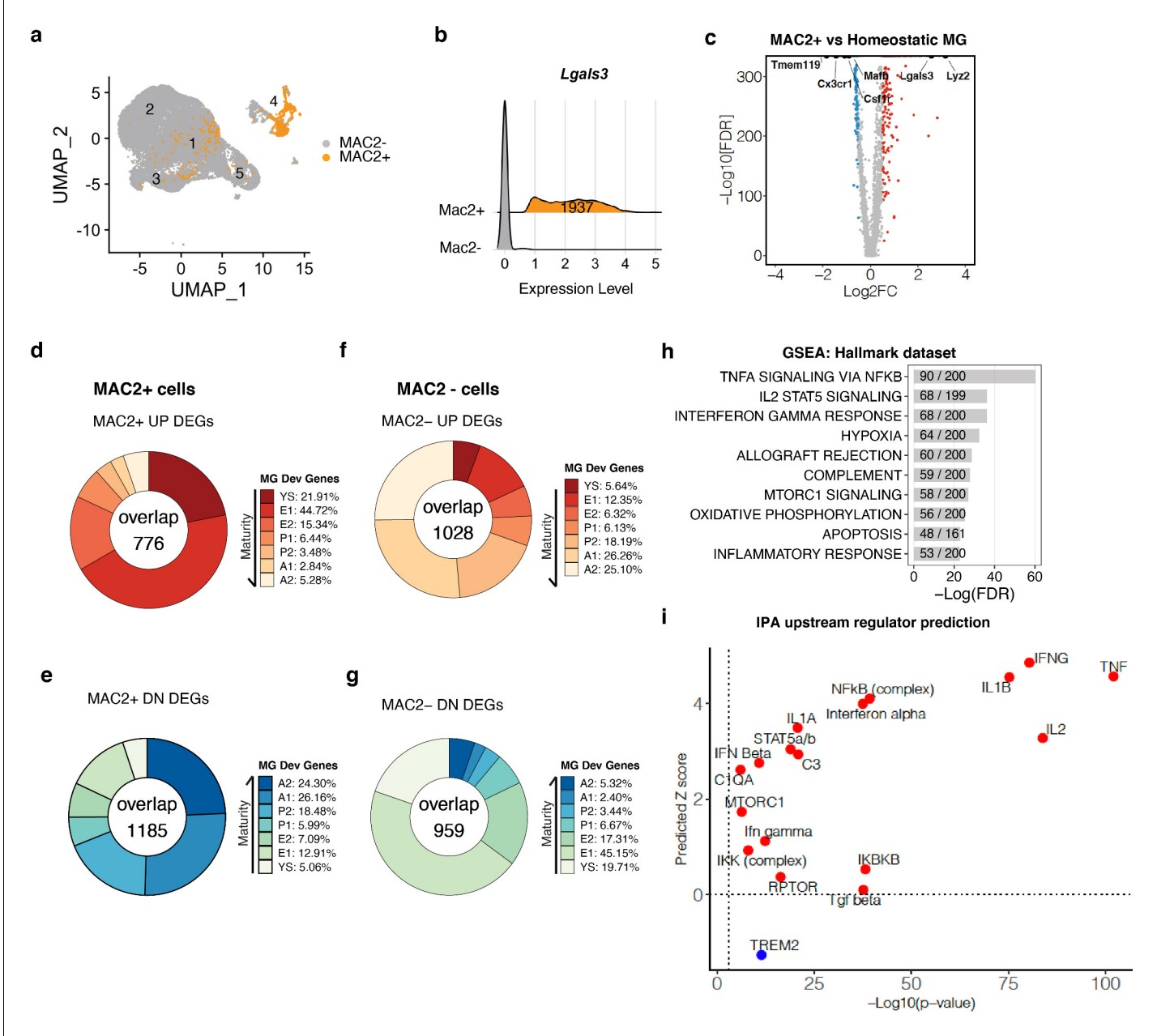

**Figure 6.** MAC2+ cells display immature microglial signatures (**a**) UMAP plot showing the spatial distribution of MAC2+ cells in different clusters. (**b**) Ridge plot showing isolation of MAC2+ cells from all clusters. MAC2+ cells were separated based on high *Lgals3* expression (mean plus one SD). (**c**) Volcano plot showing differentially expressed genes (DEGs) in MAC2+ cells compared to homeostatic microglia from Control condition (Cluster 2). Upregulated DEGs are colored in red and downregulated DEGs are colored in blue (log fold change >0.5 or <−0.5). Genes of interest are highlighted in text. (**d,e**) Donut chart showing the percentage of upregulated DEGs (**d**) and downregulated DEGs (**e**) in MAC2+ cells overlapping with developmental marker genes identified in Matcovitch-Natan and Winter et al. (**f,g**) Same analysis as in d,e, for DEGs in MAC2- cells. (**h**) Bar plot showing top-10 hallmark pathways enriched in all MAC2+ DEGs from Gene Set Enrichment Analysis (GSEA). The fraction in the bar shows the number of genes found in the DEGs (numerator) over the number of total genes curated for the corresponding pathway (denominator). (**i**) Scatterplot showing selected upstream regulators and their predicted z scores against the -Log10(p-value) based on all MAC2+ DEGs using Ingenuity Pathway Analysis (IPA).

The online version of this article includes the following source data and figure supplement(s) for figure 6:

**Source data 1.** DEGs for MAC2+ cells vs homeostatic microglia and DEGs for MAC2- vs MAC2+ cells.

**Figure supplement 1.** Microglia developmental gene signature analysis in MAC2+ cells.

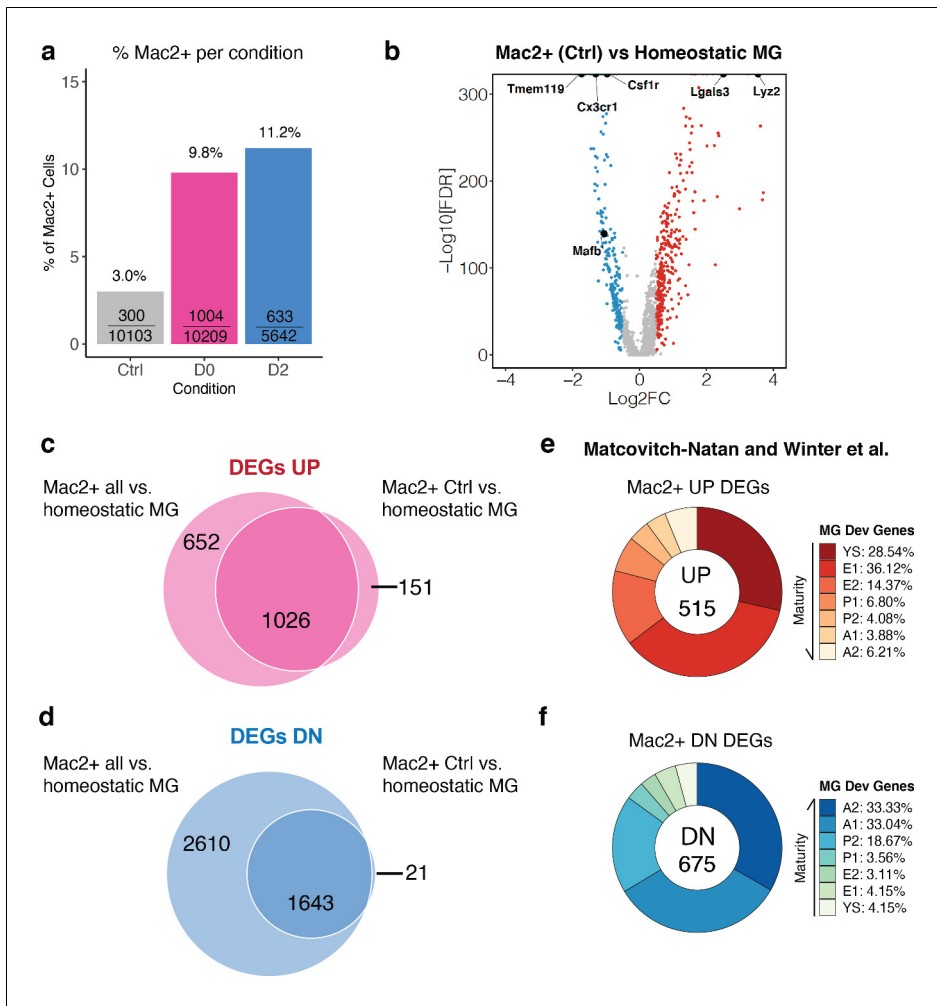

**Figure 7.** MAC2+ cells are present in naïve adult mouse brains and display immature microglial markers. (**a**) Bar graph showing the relative frequency of MAC2+ cells among all cells within each treatment group. The fraction inside each bar shows the number of MAC2+ cells within the treatment group (numerator) over the total number of cells within that treatment group (denominator). (**b**) Volcano plot showing differentially expressed genes (DEGs) of MAC2+ cells from Ctrl samples in comparison to homeostatic microglia (Cluster-2). Upregulated DEGs are colored in red while downregulated DEGs are colored in blue (log fold change >0.5 or<−0.5). Genes of interest are highlighted in text. (**c**) Venn diagram showing the common upregulated DEGs found between all MAC2+ cells (left circle) and MAC2+ cells from Ctrl samples (right circle). (**d**) Venn diagram showing the common downregulated DEGs found between all MAC2+ cells (left circle) and MAC2+ cells from Ctrl samples (right circle). (**e,f**) Donut chart showing the percentage of upregulated DEGs (**e**) and downregulated DEGs (**f**) in MAC2+ cells from Ctrl samples overlapping with developmental marker genes identified in Matcovitch-Natan and Winter et al. The online version of this article includes the following source data for figure 7:

**Source data 1.** DEGs for MAC2+ cells from control samples vs homeostatic microglia.

The existence of a CSF1R-independent microglial population in the adult CNS was suggested by previous studies. For example, regardless of the dose and duration of PLX treatment, 1–10% of the microglial population has been observed to survive (*Acharya et al., 2016*; *Rice et al., 2017*; *Huang et al., 2018*; *Zhan et al., 2019*). In *Csf1r* KO mice, although 99% of IBA1+ microglia are lost in almost all brain regions, 13.9% microglia remain in the hippocampus and 33.7% remain in the piriform cortex (*Erblich et al., 2011*). During development, although it is well known that myeloid cells require CSF1R signaling for survival, the CD45+c-kit[lo] microglial progenitors which are found in yolk sac do not start to massively express CSF1R until days post coitum (dpc) 9.0 (*Kierdorf et al., 2013*). Furthermore, expression of CSF1 ligand remains mostly quiescent until yolk sac progenitors reach

the brain rudiment at E10.5 (*Matcovitch-Natan et al., 2016*). These findings support the notion that a CSF1R-independent microglial population in the adult CNS might reflect a similar paradigm to the developmental progenitors in which alternative survival pathways are engaged.

In characterizing the remaining microglia after acute CSF1R inhibition, we recovered a subset of cells that can be distinguished by the MAC2 antigen that was also detectable under homeostatic conditions. Unlike the vast majority of the microglial population that is sensitive to CSF1R inhibitor, the MAC2+ population modestly increased in response to the drug (*Figure 4f*). The MAC2+ cells uncovered under either homeostasis (Ctrl) or under CSF1R inhibition/early repopulation (D0/D2) exhibited striking similarities with early microglial progenitors (*Figure 6*). In particular, MAFB, a transcriptional factor required for microglia maturation (*Matcovitch-Natan et al., 2016*) was significantly downregulated. Comprehensive comparison of transcriptomes revealed that MAC2+ microglia in adult brain exhibit significant overlap with those found in yolk sac and in early embryonic development, highlighting the unusual heterogeneity and plasticity of adult microglia.

MAC2, also known as Galectin-3, is a galactoside-binding protein highly expressed in myeloid cells that can be secreted to modulate a wide variety of immune functions (*Rahimian et al., 2018*). Using fate-mapping, we determined that the MAC2+ cells observed under CSF1R inhibition were not of monocytic origin, but represent a subpopulation of microglia. Expression of MAC2 is associated with an activated microglial state (*Lalancette-Hébert et al., 2012*) and has been found to promote microglial migration (*Wesley et al., 2013*). Microglia in *Mac2* knockout mice have far less proliferation in response to ischemic lesions (*Lalancette-Hébert et al., 2012*), suggesting a critical role of MAC2 in modulation of microglial proliferation. Whether deletion of *Mac2* diminishes the number of repopulating microglia following CSF1R inhibition remains to be determined. Interestingly, a more recent study showed MAC2 colocalizes with TREM2 in microglial processes, and stimulates TREM2-TYROBP signaling (*Boza-Serrano et al., 2019*). Whether activation of TREM2-TYROBP signaling is involved in the resistance of MAC2+ microglia to CSF1R inhibitor is not known.

Our findings revealed a progenitor population hidden among other microglia under steady-state conditions. Unlike other terminally differentiated myeloid cells, microglia are sufficiently self-maintained (*Ajami et al., 2007*; *Mildner et al., 2007*). Under acute microglial ablation, the 1–10% remaining microglia can restore the entire empty niche (*Huang et al., 2018*; *Zhan et al., 2019*), without needing any external progenitor input. However, due to the unusual plasticity of microglia, it still remains unclear whether a specialized adult progenitor is required to maintain microglial homeostasis. It had been proposed that every microglial cell could have the ability to conduct self-renewal when needed (*Tay et al., 2017*). Consistent with this notion, although we found that MAC2 + cells were highly proliferative during early stage microglial repopulation, a large proportion of proliferating microglia did not express MAC2. And the number of MAC2+ cells returned to Ctrl level by D14. The overall contribution of MAC2+ microglia during repopulation is hard to determine with the current approach since MAC2 expression could be very transient, and rapidly turned off once cell cycle is complete. To determine the overall contribution of MAC2+ cells to microgliogenesis, further lineage tracing studies of the descendants of MAC2+ progenitors are needed.

## Materials and methods

**Key resources table**

| Reagent type (species) or resource | Designation | Source or reference | Identifiers | Additional information |
| --- | --- | --- | --- | --- |
| Strain, strain background (*M. musculus*) | C57BL/6J | The Jackson laboratory | Jax: 000664 | |
| Genetic reagent (*M. musculus*) | *CX3CR1-CreERT2* | The Jackson laboratory | Jax: 021160 | |
| Genetic reagent (*M. musculus*) | *Rosa26-stop-DsRed* | doi: 10.1002/ eji.200636745 | MGI: 104735 | |

*Continued on next page*

*Continued*

| Reagent type (species) or resource | Designation | Source or reference | Identifiers | Additional information |
|---|---|---|---|---|
| Antibody | rabbit polyclonal anti-P2RY12 | Dr. David Julius, UCSF | N/A | 1:500 |
| Antibody | Rabbit monoclonal anti-TMEM119 | Abcam | ab209064 | 1:250 |
| Antibody | Goat polyclonal anti-IBA1 | Abcam | ab5076 | 1:500 |
| Antibody | Rat monoclonal anti-MAC2 | Cedarlane | CL8942AP | 1:1000 |
| Antibody | Goat polyclonal anti-RFP | Rockland | 200-101-379 | 1:500 |
| Antibody | Rabbit monoclonal anti-CD74 | Abcam | ab245692 | 1:500 |
| Antibody | Rabbit monoclonal anti-KI67 | Abcam | ab16667 | 1:100 |
| Antibody | Rat monoclonal APC anti-CD11b | Tonbo bioscience | 20–0112 | 1:100 |
| Antibody | Mouse monoclonal TruStain FcX (anti-mouse CD16/32) | BioLegend | 101319 | 1:50 |
| Commercial assay or kit | Chromium Single Cell 3' GEM, Library and Gel Bead Kit v3 | 10x Genomics | PN-1000075 | |
| Chemical compound, drug | SYTOX Blue Dead Cell Stain | Thermofisher | S34857 | 1:1000 |
| Chemical compound, drug | PLX5622 | Plexxikon Inc | D11100404i | 1200 mg/kg |
| Software, algorithm | Graphpad prism | Graphpad, San Diego, CA | RRID:SCR_002798 | |
| Software, algorithm | Fiji | doi: 10.1038/nmeth.2019 | RRID:SCR_002285 | |
| Software, algorithm | Cell Ranger | 10x Genomics | N/A | 3.1.0 |
| Software, algorithm | Seurat | doi: 10.1038/nbt.4096 | N/A | 3.1.2 |
| Software, Algorithm | EdgeR | doi: 10.1093/bioinformatics/btp616 doi: 10.1093/nar/gks042 | N/A | 3.28.1 |

## Mice

All animal work was performed in accordance with the Institutional Animal Care and Use Committee guidelines, at the University of California, San Francisco and at Weill Cornell Medicine, New York. Mice, with unrestricted access to water and food source, were housed in a pathogen-free barrier facility operated on a 12 hr light on/off cycle. The C57BL/6J mice were supplied by National Institute on Aging (Charles River, Wilmington, MA, USA). *CX3CR1-CreERT2/Rosa26-stop-DsRed* mice were generated by crossing the *CX3CR1-CreERT2* line (JAX: 021160) and the *Rosa26-stop-DsRed* line

(MGI: 104735). Equal numbers of male and female mice were used for all experiments except for the scRNA-seq experiments, which used only female mice.

## Drug administrations

For acute microglial ablation, mice were administered CSF1R antagonist PLX5622 orally via PLX diet (1200 mg/kg PLX5622, Plexxikon Inc, Berkeley, USA). Control diet (CD) with the same base formula was used as control. For lineage mapping, tamoxifen (Sigma-Aldrich, T5648) was dissolved in corn oil and administered to the *CX3CR1-CreERT2/Rosa26-stop-DsRed* mice by intraperitoneal (I.P.) injection for 10 days at a daily dose of 2 mg. To label proliferative cells, a solution containing 20 mg/mL 5-Ethynyl-2'-deoxyuridine (EdU) (Santa Cruz, sc-284628) was prepared fresh in sterile PBS and given to mice via I.P. injection at a dose of 80 mg/kg per animal. To maximize EdU labeling, two injections, separated by 7 hr, were given on the same day (Repopulation day 3). To detect EdU-labeled cells in brain sections, Click-iT EdU imaging kits (ThermoFisher Scientific, C10337) were used following the manufacturer's instructions before the immunofluorescence staining procedures.

## Tissue preparation for microglia isolation

Fifteen female mice (three for Ctrl, six for D0, four for D2) were perfused with PBS transcardially to remove circulating blood cells in the CNS. Whole brain was then dissected, with the cerebellum removed, and homogenized in an enzymatic digestion buffer containing 0.2% Collagenase Type 3 (Worthington, LS004182) and 3 U/mL Dispase (Worthington, LS02104). The digestion was performed at 37°C for 45 min and quenched by an inactivation buffer containing 2.5 mM EDTA (Thermofisher, 15575020) and 1% fetal bovine serum (Invitrogen, 10082147). The homogenates were kept at 4°C for all downstream applications. The homogenates were then processed for myelin depletion using myelin removal beads (Miltenyi Biotec, 130-096-733) and passed through LD column (Miltenyi Biotec, 130-042-901). The myelin-depleted fraction was then used for FACS.

## Adult microglia purification via fluorescence-activated cell sorting (FACS)

To perform immune labeling, myelin-depleted cell suspensions were incubated with TruStain fcX (BioLegend, Cat. No. 101319, clone 93) for 10 min at 4°C to block Fc receptors (1:50 dilution). To label myeloid cells, the homogenates were incubated with a Cd11b antibody conjugated with APC fluorophore for 20 min at 4°C (Tonbo bioscience, Cat: 20–0112, clone M1/70, 1:100 dilution). Sytox-blue live/dead stain (Thermofisher, S34857) was included 5 min before sorting (1:1000 dilution). Cell suspensions were then sorted on a flow cytometer (BD FACSAria II). Live Cd11b+ fraction was then sorted into pre-chilled RPMI-1640 media (Thermofisher Scientific, 12633012) containing 5% FBS. FACS gating strategy is shown in *Figure 1—figure supplement 1*. The FACS procedures were performed by the Flow Cytometry Core Facility at Weill Cornell Medicine, New York.

## Single-cell cDNA library preparation and sequencing

Single-cell gene expression profiling was performed using the 10x Genomics platform according to the manufacturer's instructions (10x Genomics, Pleasanton, CA, USA). FACS sorted cells were captured in droplets that were emulsified with gel beads containing barcoded primers (https://www.nature.com/articles/ncomms14049). Single-cell expression libraries were prepared using the 10x genomics Chromium single-cell 3' library and gel bead kit v3 reagents (10x Genomics, PN-1000075). Briefly, suspension containing 4,000–10,000 single cells were loaded into the 10x single-cell A chip and processed in the 10x Chromium controller. After reverse transcription, cDNAs were amplified by 12–14 PCR cycles. An aliquot of 10 ul cDNA product was then used for library construction, which included 14 cycles of sample index PCR to barcode samples. The quality of the cDNA library was checked by the Bioanalyzer (Agilent). KAPA qPCR was performed to measure the library quantities. A pool of all quantity-balanced libraries was then sequenced on the NovaSeq 6000 sequencer (Illumina) at an average sequencing depth of 45,889 reads/cell (post normalization). The single-cell RNA-sequencing steps were performed by the Genomics Resources Core Facility at Weill Cornell Medicine, New York.

## Bioinformatics

The Illumina BCL output files were processed by the Cell Ranger software (3.1.0, 10x Genomics). Sequencing reads were mapped to the mouse genome (mm10). A total of 36,391 cells were captured. On average, 2459 genes/cell and 7,837 UMIs/cell were detected. Over 83.8% reads were detected in cells. Downstream gene profile analyses were performed using the Seurat package in R Seurat 3.0, (*Butler et al., 2018*). The data set was filtered based on the following criteria: (1) cells with UMI count below 500 or above 20,000 were removed; (2) cells that have less than 200 genes were removed; (3) cells that have greater than 10% mitochondrial genes were removed; (*Figure 1—figure supplement 2a-d*). In addition, we also removed genes that showed expression in less than 10 cells. After data filtering and removing outliers (*Figure 1—figure supplement 4*) and contaminating (non-microglial) clusters, we obtained a total 25, 954 cells for subsequent analysis. The data were normalized by log transformation followed by regression based on total UMI counts and mitochondrial gene content. Principal component analysis was performed using the top 3000 most variable genes. Genes associated with principle components (1 to 10) were used for data dimensional reduction based on UMAP to generate distinctive cell clusters (*Figure 1—figure supplement 2e-f*). Data visualization including heatmaps, UMAP plots, and violin plots, ridge plots were generated using the following built-in functions from Seurat 3.0: 'DoHeatmap', 'FeaturePlot', 'VlnPlot', and 'RidgePlot'. Differentially expressed genes (DEGs) were performed using the 'FindMarkers' function with log fold-change threshold set at 0.25.  GLM-framework MAST was used as statistical test for the DEG analysis (*Finak et al., 2015*). Cluster annotation was performed manually with reference to the Tabula-muris dataset (*Tabula Muris Consortium et al., 2018*). Gene network analyses were performed with gene set enrichment analysis (GESA) with molecular signatures database (MSigDB) (*Mootha et al., 2003*; *Subramanian et al., 2005*). Upstream regulator prediction was done using IPA (QIAGEN Inc, https://www.qiagenbioinformatics.com/products/ingenuity-pathway-analysis). Differential abundance analysis of cell clusters was done using the negative binomial generalized linear model (NB GLM) in EdgeR to calculate p-values. Adapting the differential abundance analysis for mass cytometry data, we inputted counts as cells per label or sample instead of counts being reads per gene, and used the 'glmQLFTest' function in EdgeR (*Lun et al., 2016*; *Robinson et al., 2010*). The raw count matrix of the scRNA-seq data can be accessed on GEO with accession number (GSE150169). A list of R codes used for the analyses are available on Github (https://github.com/lifan36/Zhan-Fan-et-al-2019-scRNAseq; *Zhan, 2020*; copy archived at swh:1:rev:c008b3ff49d567a1abe9d9eed42afed4e9a27b42).

## Immunohistochemistry

Perfused mouse brains were fixed in 4% paraformaldehyde prepared in PBS for 48 hr and then incubated in 30% sucrose for at least another 48 hr. Coronal sections of 30 μm thickness were obtained by cutting the fixed brains on a sliding microtome (Leica, SM2010R). Sections were stored in cyroprotectant and kept at −20°C before use. To perform immunohistochemistry staining, sections near the same stereological position were used. One or two sections per mouse were used for each staining. Floating sections were washed in PBST buffer (PBS containing 0.5% Triton X-100) 3 times for five mins each. 3% normal donkey serum (NDS) was used for blocking at room temperature for 1 hr. Floating sections were incubated with primary antibodies diluted in PBST containing 3% NDS at 4°C overnight. The following primary antibodies and respective dilution ratio were used: P2RY12 (1:500, a gift from Dr. David Julius, University of California San Francisco); TMEM119 (1:250, Abcam, ab209064); IBA1 (1:500, Abcam, ab5076); MAC2 (1:1000, Cedarlane, CL8942AP); RFP (1:500 Rockland, 200-101-379); CD74 (1:500, Abcam, ab245692); KI67 (1:100, Abcam, ab16667). Antigen retrieval was performed for KI67 staining. For antigen retrieval, floating sections were incubated in sodium citrate buffer (10 mM sodium citrate, 0.05% Tween 20, pH 6.0) at 95°C for 30 min before blocking.

Secondary antibodies were diluted at 1:1000 in PBST containing 3% NDS. Floating sections were incubated with diluted secondary antibodies at room temperature for 1 hr with gentle shaking. All secondary antibodies used were donkey IgG conjugated with different fluorophores including, Cy3, Alexa Fluor 488, Alexa Fluor 647. All secondary antibodies were obtained from Jackson ImmunoResearch. After secondary antibody staining, sections were washed in PBST for four times, 10 min

each. DAPI was included in the last washing step as nuclei stain. Sections were then placed on glass slides and mounted with VECTASHIELD antifade mounting media (Vector Laboratories, H-1000).

## Microscopy

Immunofluorescence images of the sections were acquired on the VERSA automated slide scanner (Leica Biosystems, Wetzlar, Germany). The microscope was equipped with an Andor Zyla 5.5 sCMOS camera (Andor Technologies, Belfast, UK) and was funded under NIH S10 grant OD021717. Image acquisition was performed using the ImageScope software (Aperio Technologies, Vista, CA). The output SCN image files were processed by the Bio-Formats software (*Linkert et al., 2010*). Confocal microscopy was performed on a Zeiss LSM880 inverted scanning confocal microscope (Carl Zeiss Microscopy, Thornwood, NY). The microscope is equipped with 2 PMT detectors, a high-sensitivity GaAsP detector, and a 32-GaAsP Airyscan super resolution detector. Images were acquired using the Zeiss Zen imaging software. Z-stacks of confocal images were acquired with 8–12 focal planes at 0.75–1 μm interval. Representative images were created using max projections from Z-stacks.

## Image analyses and quantification

All image analyses were performed in FIJI V1.50i (*Schindelin et al., 2012*). Analysis macros were written using the IJM scripting. Briefly, multi-channel images were split to each individual channel. Image segmentation was performed using the adaptive threshold approach (https://sites.google.com/site/qingzongtseng/adaptivethreshold). Cells showing two different markers (e.g. IBA1+ and MAC2+) were selected if the cell mask from each channel showed greater than 10% overlap. Quantification including cell counting and area measurement was performed using the 'Analyze Particles' and 'ROI Manager' function in FIJI. A list of the analysis IJM code used in the study are available on Github (https://github.com/lifan36/Zhan-Fan-et-al-2019-scRNAseq).

## Statistics

All experiments were performed with a minimum of at least three biological replicates. Mean values from each animal were used for computing statistical differences. Standard error of the mean (SEM) was used for error bars. Statistical analyses were performed in Graphpad prism 8.0 (Graphpad, San Diego, CA) and R (F Foundation for Statistical Computing, Vienna, Austria). Data visualization was achieved with R package ggplot2 (*Wickham, 2009*). Data normality was assessed using the Shapiro-Wilk normality test. F test was used to assess homoscedasticity prior to unpaired t-test and Brown-Forsythe test was used to homoscedasticity prior to ANOVA. For data with normal distribution and equal variance, unpaired t-test was used to compare two groups. One-way ANOVA was used to compare data with more than two groups. Dunnett's multiple comparisons test was used to compare difference between designated groups. Two-way ANOVA with Dunnett's multiple comparisons test was used for multiple group comparison to Ctrl. For data that failed to pass the normality test, Mann Whitney test was applied. For data with unequal variance, unpaired t-test with Welch's correction was applied. p-value and FDR are summarized as ns ($p > 0.05$); * ($p \leq 0.05$); ** ($p \leq 0.01$); *** ($p \leq 0.001$); **** ($p \leq 0.0001$).

## Acknowledgements

We want to thank Dr. David Julius from University of California San Francisco for the P2ry12 antibody; Parmveer Singh from Plexxikon Inc, for supplying the PLX diet; Zlata Plotnikova for assistance with the microglia isolation; Jason McCormick and Tomas Baumgartner from Weill Cornell Medicine Flow Cytometry Core Facility for FACS assistance; Dong Xu, Xing Wang and Adrian Tan from Genomics Resources Core Facility for performing single-cell RNA sequencing; Rashad Ahmed from Scientific Computing Unit for assistance on cluster computing; Dr. Matthew Lee Settles from University of California-Davis for the scRNA-seq analysis training course; Cody Jackson from the Gladstone Histology and Light Microscopy Core for imaging assistance; and Dr. Kathryn Claiborn for editing the manuscript. NIH S10 RR028962 funded the use of FACSAria cell sorter. This work was supported by NIH grants 1R01AG054214-01A1, U54NS100717, R01AG051390, and Tau Consortium grant, and JPB Foundation (to LG), the National Institute of Aging Grant F30 AG062043-02 and National Institute of Health Grant T32GM007618 (to LK).

## Additional information

### Competing interests
Li Gan: Li Gan is a founder of Aeton Therapeutics, Inc. The other authors declare that no competing interests exist.

### Funding

| Funder | Grant reference number | Author |
| --- | --- | --- |
| National Institutes of Health | 1R01AG054214-01A1 | Li Gan |
| National Institutes of Health | U54NS100717 | Li Gan |
| National Institutes of Health | R01AG051390 | Li Gan |
| National Institutes of Health | Tau Consortium grant | Li Gan |
| National Institute of Aging | F30 AG062043-02 | Lay Kodama |
| National Institutes of Health | T32GM007618 | Lay Kodama |

The funders had no role in study design, data collection and interpretation, or the decision to submit the work for publication.

### Author contributions
Lihong Zhan, Conceptualization, Data curation, Software, Formal analysis, Validation, Investigation, Visualization, Methodology, Writing - original draft, Writing - review and editing; Li Fan, Lay Kodama, Data curation, Software, Formal analysis, Investigation, Visualization, Methodology, Writing - original draft, Writing - review and editing; Peter Dongmin Sohn, Project administration, Writing - review and editing; Man Ying Wong, Investigation, Methodology; Gergey Alzaem Mousa, Formal analysis, Validation, Writing - review and editing; Yungui Zhou, Yaqiao Li, Data curation, Formal analysis; Li Gan, Conceptualization, Resources, Supervision, Funding acquisition, Investigation, Writing - original draft, Writing - review and editing

### Author ORCIDs
Li Fan (iD) https://orcid.org/0000-0003-1780-6919
Li Gan (iD) https://orcid.org/0000-0003-4600-275X

### Ethics
Animal experimentation: All of the animals were handled according to approved institutional animal care and use committee (IACUC) protocols (#AN173162-02) of the University of California, San Francisco.

### Decision letter and Author response
Decision letter https://doi.org/10.7554/eLife.51796.sa1
Author response https://doi.org/10.7554/eLife.51796.sa2

## Additional files

### Supplementary files
• Transparent reporting form

### Data availability
The raw count matrix of the scRNA-seq data can be accessed on GEO with accession number (GSE150169). A list of R codes used for the analyses are available on Github (https://github.com/lifan36/Zhan-Fan-et-al-2019-scRNAseq; copy archived at https://archive.softwareheritage.org/swh:1:rev:c008b3ff49d567a1abe9d9eed42afed4e9a27b42).

The following dataset was generated:

| Author(s) | Year | Dataset title | Dataset URL | Database and Identifier |
|---|---|---|---|---|
| Zhan L, Gan L, Fan L | 2020 | scRNA-seq | http://www.ncbi.nlm.nih.gov/geo/query/acc.cgi?acc=GSE150169 | NCBI Gene Expression Omnibus, GSE150169 |

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
