## [Decision Letter]

**Acceptance summary:**

This study addresses an important question in the field: What is the identify and ontogeny of the small percentage of Csf1 receptor- independent population of resident microglia that repopulate the brain following csf1r inhibition or loss of function? Here the authors applied single-cell RNA sequencing to examine the remaining population of microglia following acute csf1r inhibition with PLX. Several populations or states of microglia were identified, including a subpopulation that express Mac2/Galectin2. Lineage tracing and other approaches provide evidence that Mac2+ are proliferative and not derived from peripheral monocytes and that this population express microglia progenitor signatures that are normally present as part of the homeostatic population in the adult brain.

**Decision letter after peer review:**

Thank you for submitting your article "A Mac2-positive progenitor-like microglial population survives independent of CSF1R signaling in adult mouse brain" for consideration by *eLife*. Your article has been reviewed by three peer reviewers, and the evaluation has been overseen by a Reviewing Editor and Marianne Bronner as the Senior Editor. The following individual involved in review of your submission has agreed to reveal their identity: Frederick Christian Bennett (Reviewer #2).

The reviewers have discussed the reviews with one another and the Reviewing Editor has drafted this decision to help you prepare a revised submission.

Summary:

This is an interesting study that addresses an important question in the field: what is different about microglia that survive in the presence of the Csf1r inhibitor PLX5622 compared to those that are eliminated? Here the authors applied single-cell RNA sequencing to examine the remaining population of microglia following acute csf1r inhibition with PLX. Several populations or states of microglia were identified, including a subpopulation that express Mac2/Galectin2. They carried out lineage tracing experiments (using Cx3CR1-CreErT2-RFP) and provide evidence that Mac2+ are proliferative and not derived from peripheral monocytes and that this population express microglia progenitor signatures that are normally present as part of the homeostatic population in control mice.

Though limited by a small number of sequenced cells and few biological replicates, the study presents interesting data about "inhibitor resistant" and repopulating microglia. However, there are several controls and issues related to RNA-seq analyses (validation, biological variability and reproducibility) that must be addressed to support their conclusions. Clarity is needed on data analyses and experimental design, including pseudotime analyses, as summarized below. For example, it is not clear whether clusters 4,5, and 8 are thought by the authors to be microglia, just ones that express some monocyte/granulocyte/DC genes in setting of plx5622 treatment. This is an important point to address in order to interpret the data, and the number of microglia sequenced. If these clusters are truly not microglia, then it would likely not be possible to perform the comparison between Mac2+ microglia Mac2- microglia. If they are microglia, it would be important for the authors to compellingly show this (either by analyzing current data, or showing staining of parenchymal microglia expressing some of these marker). Finally, please address or modify the claim that remaining microglia following PLX5622. treatment are "Csf1r independent," which is not supported by the data.

Essential revisions:

1) RNA sequencing experiments: The authors do not provide quantification of biological variability nor justification across 3 conditions.

– The most problematic is that the single-cell is based off what is effectively an n=1 for each group (the D0 group had two animals but they were pooled together). This experimental design leaves no room for interpretation of any potential outliers or technical artifacts in the data. Figure 1F-N is uninterpretable as relative frequency comparisons with n=1 are not valid. Figure S2A-C are also important comparisons which need statistical comparison that is not possible with n=1. Figure 7 also cannot be interpreted due to this issue.

n=3 replicates (2 additional replicates per group) should be performed and combined with the original analysis. DEG based analysis also significantly suffers from same issue as DEG between replicates cannot be analyzed.

2) Please provide clarity and address issues related to Figures 3 and 5 and 6:

a) Figure 3: The authors analyze gene expression across cell clusters, and conclude that many clusters express genes characteristic of monocytes, granulocytes, NK cells, T cells, and dendritic cells. Supplementary data shows that NK and T cells are not found in the brain parenchyma. The authors claim that these cells must have been unintentionally captured in their CD11b+ gate, but do not provide analogous validation of the "monocyte-like" "granulocyte-like," and " dendritic cell-like" clusters as being either present or absent from the parenchyma. I infer that the authors are trying to suggest that these "-like," (eg, "granulocyte-like") populations are real microglia that have adopted alternative gene expression patterns due to PLX5622 treatment, but this is not made clear. In-situ validation showing parenchymal microglia to express genes in these clusters would be useful but not necessary. More critically, the manuscript would be easier to interpret if the authors explicitly stated whether they believe these cells to be microglia or not, and what makes them think so from the data. As it stands, this grey area around the identity of clusters 4,5, and 8 complicates interpretation.

b) The use of pseudotime analysis in 3F/G is unclear. The T and NK cells sequenced in this study are almost certainly not on a temporal continuum of cell states with the other clusters, so trying to make inferences about them as such is of unclear validity. The authors could more clearly explain the purpose of applying pseudotime here, and their interpretation of its biological meaning given the above, or remove this analysis. Clusters 4-8 (and even cluster 3) are not in the same trajectory as microglia as they are completely different cell types. This analysis should be redone using only microglia or removed completely.

c) Figure 6: Analysis of DEG of Mac2+ cells is similarly problematic. The authors perform the analysis on all Mac2+ cells. However, the vast majority of these cells are non-microglial cells. It does not make sense include non-microglial cells in an analysis compared to homeostatic microglia. This analysis should only be performed between Mac2+ microglia Mac2- microglia. It would appear they may not be enough microglia to make this comparison in the current dataset but this only further underlines the importance of this issue.

3) Lineage tracing experiment Figure 5A-D: The authors state that there is no significant difference in RFP+ Mac2+ cells so they don't appear to be peripherally derived. However, they fail to address fact that only 80% or less of the Mac2+ cells are RFP+. Please provide quantification of microglial labeling efficiency to control for this to support this conclusion.

4) Csf1r independence: The claim (including in the title) that Mac2+ microglia survive independent of CSF1r signaling is not supported by the presented data. Using both scRNA-seq and immunolabeling, the authors convincingly demonstrate a small subset of microglia to be Mac2+. During treatment with PLX5622, this Mac2+ subset expands modestly, and the authors conclude that there is a subset of *Csf1r*-independent Mac2+ microglia. The interpretation that microglia surviving PLX5622 treatment, in particular Mac2+ cells, are "*Csf1r*-independent," is not supported by the presented data, as there are many alternative explanations for why they survive. Proving Csf1r independence experimentally would require a large amount of additional work, but the authors could alternatively modify their language and claims to reflect that these cells are relatively resistant to inhibitor treatment, rather than Csf1r independent. This is a very important distinction for future studies focused on this inhibitor resilient population, as the mechanism of survival remains unknown.

[Editors' note: further revisions were suggested prior to acceptance, as described below.]

Thank you for resubmitting your article "A Mac2-positive progenitor-like microglial population is resistant to CSF1R inhibition in adult mouse brain" for consideration by *eLife*. Your revised article has been reviewed by two peer reviewers, and the evaluation has been overseen by a Reviewing Editors and Marianne Bronner as the Senior Editor. The following individual involved in review of your submission has agreed to reveal their identity: F. Chris Bennett (Reviewer #4).

The reviewers have discussed the reviews with one another and the Reviewing Editor has drafted this decision to help you prepare a revised submission.

Summary:

This is an interesting study that addresses an important question in the field: what is different about microglia that survive in the presence of the Csf1r inhibitor PLX5622 compared to those that are eliminated? Here the authors applied single-cell RNA sequencing to examine the remaining population of microglia following acute csf1r inhibition with PLX. Several populations or states of microglia were identified, including a subpopulation that express Mac2/Galectin2. They carried out lineage tracing experiments and provide evidence that Mac2+ are proliferative and not derived from peripheral monocytes and that this population express microglia progenitor signatures that are normally present as part of the homeostatic population in control mice.

Though limited by a small number of sequenced cells and few biological replicates, the study presents interesting data about "inhibitor resistant" and repopulating microglia. However, there are several controls and several issues, related to RNA-seq analyses (validation of key findings, biological variability and reproducibility) that must be addressed to support their conclusions as summarized below.

The authors have made significant improvements and clarifications in the analysis in their revised paper. However, one key issue has arisen that needs to be addressed regarding the annotation of cluster 5 and 6 (but especially cluster 6). They are defined in the paper as DC-like microglia and anti-microbial microglia respectively. Cluster 6 (and 5) are not likely microglia and don't appear to be myeloid in origin and are almost certainly granulocytes. Please address this remaining issue by providing evidence that these populations are not contaminating cells, or please exclude those cells from analysis as summarized below (Reviewer 1). If the authors still find that Lgals3 is specific marker of repopulated in clusters 1-4 then their main conclusion holds. However, as much of the analyses of differential gene expression analysis depends on these clusters, this analysis will also need to be redone with this in mind.

Revisions expected in follow-up work:

See reviewer 1 and 2 comments below.

Reviewer #1:

I would like to thank the authors for their substantive improvements and clarifications in the analysis in their revised paper that is much improved. However, one key issue has arisen that needs to be addressed regarding the annotation of cluster 5 and 6 but especially cluster 6.

Main concerns:

1) There are significant problems with the annotations of clusters 5 and cluster 6. They are defined in the paper as DC-like microglia and anti-microbial microglia respectively.

– Cluster 6 are almost certainly not microglia and don't appear to be myeloid in origin and are almost certainly granulocytes. As seen in SI Figure 2 and Figure 3 cluster 6 has little to no expression of P2ry12, P2ry13, Cx3cr1, Trem2, Fcrls, Aif1, or Tmem119. Furthermore, they are enriched for classic neutrophil markers S100a9, S100a8, Ngp.

– Cluster 5 are slightly trickier with the markers presented but based on the literature seem more akin to brain associated macrophages/monocytes (perivascular and meningeal) than they do to microglia, but it remains possible they represent an early precursor myeloid population. Again, as seen in SI Figure 2 and Figure 3, they have little to no expression of classic microglia or myeloid markers.

Both of these annotations impact for the conclusions of the paper as they are the clusters expressing the majority of Mac2+ cells. Thus, defining these clusters as microglia or microglia progenitors needs further validation.

Currently the authors show that Mac2+ Iba1+ cells are present by IHC and that CD74/Iba1+ cells are present via IHC. Unfortunately, neither of these confirms that these represent cells from clusters 5 and 6. There are as the authors show in SI Figure 6A Mac2+ cells in cluster 1-4 (which are microglia/myeloid) and given the lack of expression of Iba1 or other myeloid markers in clusters 5-6 it seems much more likely that the cells the authors are identifying are Mac2+ cells from cluster 1-4. Furthermore CD74/Iba1 IHC images also seem more likely to represent cells in clusters 1 or 4 as those clusters also show some expression of CD74 (SI Figure 2) while cluster 5 has very low to minimal expression of Iba1.

If the authors want to show that either of these populations are indeed microglia then more validation/experiments are necessary.

a) The authors will need to show either smFISH or combined IHC/smFISH of Iba1 with S100a8, S100a9, and/or Ngp in order to demonstrate that these cells are potentially myeloid and not granulocytes and that those numbers are similar to total numbers of Iba1/Mac2+ cells.

b) The same holds true for cluster 5. We suggest the authors use MHC genes as shown in SI Figure 2 to colocalize with Iba1 as those genes appear to be more cluster specific than CD74.

Both of these impacts the conclusions that Lgals3 is unique marker of microglia progenitor as cluster 6 seem very unlikely to be microglia and cluster 5 seem potentially possible although certainly not definitive at this stage. Therefore, when the authors perform DEG analysis between Mac2+ and Mac2- cells there are of course going to be significant results as the authors are likely comparing non-microglial cells to microglia.

However, the Mac2+/Iba1+ cells are still shown to be enriched in D0 brain by IHC that much is very clear. Therefore, the Mac2+ cells in clusters 1-4 may still be of high importance during repopulation. It is even possible that once cluster 5-6 are separated from the analysis that these Mac2+ cells that can be seen in clusters 1-4 (Si Figure 6A) will even form their own cluster in the analysis.

a) The DE analysis and clustering will need to be reperformed pending the results of smFISH and other confirmation as to the origins of cluster 5-6 (see above).

Reviewer #4:

The authors have addressed the major concerns raised in the prior submission with additional data, clarified analyses, and text edits. Clarity, interpretability, and the strength of conclusions have greatly improved.

Concrete points:

1) It would be good to show some kind of error estimate for panel 1F (which might require normalizing to each replicate and not each treatment group), and this reviewer is not clear on how edgeR would be used to calculate a p value for this ratio, as is stated in the legend, or otherwise how what statistical test was used here?

2) It would be useful for interpretability of Figure 6A to see the absolute # of MAC2 pos cells in the homeostatic vs repopulating clusters (or to refer to the panel where this was calculated, in the case that I missed this).

Subjective points:

1) Language around Csf1r independence: The authors very effectively refined language around whether residual microglia are "Csf1r independent" or not. There is one sentence in the Abstract that still states that "a small subset of microglia in mouse brains can survive without Csf1r signaling", this has not been proven in prior studies, including the Pollard study in Csf1r KO mice, because this study did not test whether the IBA1 pos cells observed actually "survived" or were transient (for example, monocytes engrafting the brain and undergoing apoptosis).

2) "Progenitor signatures": The authors use the word "progenitor" to describe the young/embryonic microglial signatures used in comparisons in Figure 6 and supplement from the Matcovitch-Natan study, and describe MAC2+ cells as having "progenitor signatures." Strictly speaking, it is my impression that "progenitors" only refers to the "YS" population, not the E1 population and thereafter (which are no longer progenitors, but instead "early embryonic microglia"). They also claim that "progenitor-like" cells are present in the homeostatic brain. I would suggest changing the language to reflect the fact that microglia are not thought to have progenitors, and rather to self renew, acting as their own progenitor. Again, this is minor, I raise it only with goal of increasing precision of language since the subject of microglial progenitors is contentious, and in lumping together true yolk sac progenitors with early microglia, this may cloud the debate.

[Editors' note: further revisions were suggested prior to acceptance, as described below.]

Thank you for submitting your article "A MAC2-positive progenitor-like microglial population is resistant to CSF1R inhibition in adult mouse brain" for consideration by *eLife*. Your article has been reviewed by one peer reviewer, and the evaluation has been overseen by a Reviewing Editor and Marianne Bronner as the Senior Editor. The reviewers have opted to remain anonymous.

The reviewers have discussed the reviews with one another and the Reviewing Editor has drafted this decision to help you prepare a revised submission.

The authors have addressed nearly all stated concerns raised by prior reviewers and significantly improved this manuscript. What remains unaddressed is definitive verification that cluster 5 cells (where the great majority of Mac2+ cells reside) are parenchymal microglia as opposed to border macrophages or something else. I think this remains a substantive concern as the crux of this paper as written is that Mac2+ microglia are resistant to CSF1R inhibition, and this conclusion seems to depend on whether cluster 5 cells are microglia. Some of these cells do express Cx3cr1 and Iba, and many CD74, but these markers are unfortunately not specific to microglia.

Revisions expected in follow-up work:

1) What remains unaddressed is definitive verification that cluster 5 cells (where the great majority of Mac2+ cells reside) are parenchymal microglia as opposed to border macrophages or something else. This is a a key point to many interpretations and conclusions is that the cells in cluster 5 are indeed microglia, since they are the predominant source of mac2/lgals3 pos cells. An alternative to showing this would be to assume they are not microglia and exclude them from analyses of repopulation microglia, but the authors did not take this route. Additionally, there is no significant difference in the sequencing data between Ctrl, D0, and D2 in abundance of Cluster 5 (Figure 1F). In fact based on that figure it appears that one mouse is a significant outlier for that cluster. Please address.

2) When the authors remove clusters of contaminating cells from the analysis, it appears that they did not re-cluster the data. The UMAP plots are the same but without the contaminating clusters. This is not a valid analysis as the original clustering is based on variable genes and variable PCs from data that included all of the cells. Please address.

Reviewer #4:

The authors have addressed nearly all stated concerns raised by prior reviewers and significantly improved this manuscript. What remains unaddressed is definitive verification that cluster 5 cells (where the great majority of Mac2+ cells reside) are parenchymal microglia as opposed to border macrophages or something else. I think this remains a substantive concern as the crux of this paper as written is that Mac2+ microglia are resistant to CSF1R inhibition, and this conclusion seems to depend on whether cluster 5 cells are microglia. Some of these cells do express Cx3cr1 and Iba, and many CD74, but these markers are unfortunately not specific to microglia.

---

## [Author Response]

Though limited by a small number of sequenced cells and few biological replicates, the study presents interesting data about "inhibitor resistant" and repopulating microglia. However, there are several controls and issues related to RNA-seq analyses (validation, biological variability and reproducibility) that must be addressed to support their conclusions. Clarity is needed on data analyses and experimental design, including pseudotime analyses, as summarized below. For example, it is not clear whether clusters 4,5, and 8 are thought by the authors to be microglia, just ones that express some monocyte/granulocyte/DC genes in setting of plx5622 treatment. This is an important point to address in order to interpret the data, and the number of microglia sequenced. If these clusters are truly not microglia, then it would likely not be possible to perform the comparison between Mac2+ microglia Mac2- microglia. If they are microglia, it would be important for the authors to compellingly show this (either by analyzing current data, or showing staining of parenchymal microglia expressing some of these marker). Finally, please address or modify the claim that remaining microglia following PLX5622. treatment are "Csf1r independent," which is not supported by the data.

We thank the reviewers for the supportive comments. We have performed extensive new experiments that include 15 mice (n=3 independent samples for each condition), and expanded markedly the number of cells sequenced (a total of 30,430 cells passed QC). To increase clarity, the revised manuscript focuses on the sub-clusters that express microglial markers. We have also modified the statement to precisely describe the remaining microglial population under CSF1r inhibition.

Essential revisions:1) RNA sequencing experiments: The authors do not provide quantification of biological variability nor justification across 3 conditions.– The most problematic is that the single-cell is based off what is effectively an n=1 for each group (the D0 group had two animals but they were pooled together). This experimental design leaves no room for interpretation of any potential outliers or technical artifacts in the data. Figure 1F-N is uninterpretable as relative frequency comparisons with n=1 are not valid. Figure S2A-C are also important comparisons which need statistical comparison that is not possible with n=1. Figure 7 also cannot be interpreted due to this issue.n=3 replicates (2 additional replicates per group) should be performed and combined with the original analysis. DEG based analysis also significantly suffers from same issue as DEG between replicates cannot be analyzed.

We completely agree, and have now performed new experiments in 15 mice to achieve n = 3 per condition, including three age-matched non-treated mice (Ctrl), six C57/BL6J mice treated with PLX diet (D0, 2 brains/sample), and six mice that were switched to a control diet for 2 days after PLX treatment (D2, 2 brains/sample). Statistical comparisons in revised Figure 1 were performed using differential abundance analysis using the R package, EdgeR, which employs a generalized linear regression model(Lun et al., 2016, McCarthy et al., 2012, Robinson et al., 2010). We used the glmQLFTest function to calculate significance. Details in the Materials and methods section of the updated manuscript.

2) Please provide clarity and address issues related to Figures 3 and 5 and 6:a) Figure 3: The authors analyze gene expression across cell clusters, and conclude that many clusters express genes characteristic of monocytes, granulocytes, NK cells, T cells, and dendritic cells. Supplementary data shows that NK and T cells are not found in the brain parenchyma. The authors claim that these cells must have been unintentionally captured in their CD11b+ gate, but do not provide analogous validation of the "monocyte-like" "granulocyte-like," and " dendritic cell-like" clusters as being either present or absent from the parenchyma. I infer that the authors are trying to suggest that these "-like," (eg, "granulocyte-like") populations are real microglia that have adopted alternative gene expression patterns due to PLX5622 treatment, but this is not made clear. In-situ validation showing parenchymal microglia to express genes in these clusters would be useful but not necessary. More critically, the manuscript would be easier to interpret if the authors explicitly stated whether they believe these cells to be microglia or not, and what makes them think so from the data. As it stands, this grey area around the identity of clusters 4,5, and 8 complicates interpretation.

We thank the reviewers for the insightful comments. In the revised manuscript, we addressed these concerns in the following ways: 1) We first established the clusters that express microglial markers, and focus the analyses on those markers only. 2) For the microglial subclusters that express different markers, we now characterized them by their functionality, instead of using cell lineage characterization, based on the top DEGs in these clusters. We believe that the new characterization, and the presence of microglial markers, remove the ambiguity in the interpretation of these subclusters in different conditions.

b) The use of pseudotime analysis in 3F/G is unclear. The T and NK cells sequenced in this study are almost certainly not on a temporal continuum of cell states with the other clusters, so trying to make inferences about them as such is of unclear validity. The authors could more clearly explain the purpose of applying pseudotime here, and their interpretation of its biological meaning given the above, or remove this analysis. Clusters 4-8 (and even cluster 3) are not in the same trajectory as microglia as they are completely different cell types. This analysis should be redone using only microglia or removed completely.

We agree with the reviewer and have removed the pseudotime analyses in the revision.

c) Figure 6: Analysis of DEG of Mac2+ cells is similarly problematic. The authors perform the analysis on all Mac2+ cells. However, the vast majority of these cells are non-microglial cells. It does not make sense include non-microglial cells in an analysis compared to homeostatic microglia. This analysis should only be performed between Mac2+ microglia Mac2- microglia. It would appear they may not be enough microglia to make this comparison in the current dataset but this only further underlines the importance of this issue.

We completely agree and have performed the analyses of Mac2+ cells using those from microglial clusters only, to compare with homeostatic microglia. The new analyses are now in revised Figure 6 and 7.

3) Lineage tracing experiment Figure 5A-D: The authors state that there is no significant difference in RFP+ Mac2+ cells so they don't appear to be peripherally derived. However, they fail to address fact that only 80% or less of the Mac2+ cells are RFP+. Please provide quantification of microglial labeling efficiency to control for this to support this conclusion.

The reviewers comment prompted us to explain the methodology more explicitly. The key comparison is the labeling efficiency at baseline vs. that after repopulation, regardless of the exact labeling level. Specifically, if Mac2+ cells are derived from monocytic origin, the frequency (baseline at 80%) will decline because monocyte are not RFP+.

4) Csf1r independence: The claim (including in the title) that Mac2+ microglia survive independent of CSF1r signaling is not supported by the presented data. Using both scRNA-seq and immunolabeling, the authors convincingly demonstrate a small subset of microglia to be Mac2+. During treatment with PLX5622, this Mac2+ subset expands modestly, and the authors conclude that there is a subset of Csf1r-independent Mac2+ microglia. The interpretation that microglia surviving PLX5622 treatment, in particular Mac2+ cells, are "Csf1r-independent," is not supported by the presented data, as there are many alternative explanations for why they survive. Proving Csf1r independence experimentally would require a large amount of additional work, but the authors could alternatively modify their language and claims to reflect that these cells are relatively resistant to inhibitor treatment, rather than Csf1r independent. This is a very important distinction for future studies focused on this inhibitor resilient population, as the mechanism of survival remains unknown.

We have rephrased the statement to indicate the population to be relatively resistant, rather than independent.

[Editors' note: further revisions were suggested prior to acceptance, as described below.]

Revisions for this paper:The authors have made significant improvements and clarifications in the analysis in their revised paper. However, one key issue has arisen that needs to be addressed regarding the annotation of cluster 5 and 6 (but especially cluster 6). They are defined in the paper as DC-like microglia and anti-microbial microglia respectively. Cluster 6 (and 5) are not likely microglia and don't appear to be myeloid in origin and are almost certainly granulocytes. Please address this remaining issue by providing evidence that these populations are not contaminating cells, or please exclude those cells from analysis as summarized below (Reviewer 1). If the authors still find that Lgals3 is specific marker of repopulated in clusters 1-4 then their main conclusion holds. However, as much of the analyses of differential gene expression analysis depends on these clusters, this analysis will also need to be redone with this in mind.

The reviewers comments prompted us to examine the markers for clusters 5 and 6 more carefully. Since cluster 6 cells express very low levels of *Iba1*, we agree with the reviewer that cluster 6 might not be of myeloid origin, thus have excluded it from additional downstream analyses. However, cluster 5 cells express appreciable levels of *Iba1* (Figure 1E) and *Cx3cr1* (Figure 2—figure supplement 1). So we believe cluster 5 is more likely to be of myeloid origin, thus have included it in our subsequent analyses.

Revisions expected in follow-up work:See reviewer 1 and 2 comments below.Reviewer #1:I would like to thank the authors for their substantive improvements and clarifications in the analysis in their revised paper that is much improved. However, one key issue has arisen that needs to be addressed regarding the annotation of cluster 5 and 6 but especially cluster 6.Main concerns:1) There are significant problems with the annotations of clusters 5 and cluster 6. They are defined in the paper as DC-like microglia and anti-microbial microglia respectively.– Cluster 6 are almost certainly not microglia and don't appear to be myeloid in origin and are almost certainly granulocytes. As seen in SI Figure 2 and Figure 3 cluster 6 has little to no expression of P2ry12, P2ry13, Cx3cr1, Trem2, Fcrls, Aif1, or Tmem119. Furthermore, they are enriched for classic neutrophil markers S100a9, S100a8, Ngp.

We thank the reviewer for the insightful comment and have excluded Cluster 6 in the revised analyses.

– Cluster 5 are slightly trickier with the markers presented but based on the literature seem more akin to brain associated macrophages/monocytes (perivascular and meningeal) than they do to microglia, but it remains possible they represent an early precursor myeloid population. Again, as seen in SI Figure 2 and Figure 3, they have little to no expression of classic microglia or myeloid markers.

We agree that Cluster 5 cells are distinct compared with the homeostatic microglial populations. Indeed, one of the important features of the immature, repopulating microglial cells is the downregulation of homeostatic microglial markers. However, they do express *Iba1* and *Cx3cr1*, suggesting that they are most likely of myeloid lineage. Therefore, cluster 5 is included in the revised analyses.

Both of these annotations impact for the conclusions of the paper as they are the clusters expressing the majority of Mac2+ cells. Thus, defining these clusters as microglia or microglia progenitors needs further validation.Currently the authors show that Mac2+ Iba1+ cells are present by IHC and that CD74/Iba1+ cells are present via IHC. Unfortunately, neither of these confirms that these represent cells from clusters 5 and 6. There are as the authors show in SI Figure 6A Mac2+ cells in cluster 1-4 (which are microglia/myeloid) and given the lack of expression of Iba1 or other myeloid markers in clusters 5-6 it seems much more likely that the cells the authors are identifying are Mac2+ cells from cluster 1-4. Furthermore CD74/Iba1 IHC images also seem more likely to represent cells in clusters 1 or 4 as those clusters also show some expression of CD74 (SI Figure 2) while cluster 5 has very low to minimal expression of Iba1.If the authors want to show that either of these populations are indeed microglia then more validation/experiments are necessary.a) The authors will need to show either smFISH or combined IHC/smFISH of Iba1 with S100a8, S100a9, and/or Ngp in order to demonstrate that these cells are potentially myeloid and not granulocytes and that those numbers are similar to total numbers of Iba1/Mac2+ cells.

Cluster 6 was removed for data analysis.

b) The same holds true for cluster 5. We suggest the authors use MHC genes as shown in SI Figure 2 to colocalize with Iba1 as those genes appear to be more cluster specific than CD74.

To assess if MHC genes are better markers for cluster 5, we directly compared the expression of *CD74* with that of MHC genes in cluster 5 as ridge plots (Author response image 1). Although expression of *H2-Eb1, H2-Aa, H2-Ab1* are restricted to cluster 5, levels of MHC genes are lower than that of *CD74*. Thus, CD74 immunostaining offers a better validation than MHC genes.

In the revised Figure 2E, we included a ridge plot showing that the expression levels of CD74 is much higher in cluster 5 than in the other clusters, providing direct evidence that CD74 is a highly specific marker for cluster 5.

**Author response image 1. sa2fig1:** Ridge plot shows Comparison of expression levels of CD74 and MHCs in clusters1–5. Cells in cluster 5 have the highest levels of CD74, H2-Eb1, H2-Aa and H2-Ab1. Among them, CD74 exhibit the highest level.

Both of these impacts the conclusions that Lgals3 is unique marker of microglia progenitor as cluster 6 seem very unlikely to be microglia and cluster 5 seem potentially possible although certainly not definitive at this stage. Therefore, when the authors perform DEG analysis between Mac2+ and Mac2- cells there are of course going to be significant results as the authors are likely comparing non-microglial cells to microglia.However, the Mac2+/Iba1+ cells are still shown to be enriched in D0 brain by IHC that much is very clear. Therefore, the Mac2+ cells in clusters 1-4 may still be of high importance during repopulation. It is even possible that once cluster 5-6 are separated from the analysis that these Mac2+ cells that can be seen in clusters 1-4 (Si Figure 6A) will even form their own cluster in the analysis.a) The DE analysis and clustering will need to be reperformed pending the results of smFISH and other confirmation as to the origins of cluster 5-6 (see above).

The DE analysis between clusters for Figures 6 and 7 were reperformed after the removal of cluster 6. The results are very similar and the conclusions remain unchanged.

Reviewer #4:The authors have addressed the major concerns raised in the prior submission with additional data, clarified analyses, and text edits. Clarity, interpretability, and the strength of conclusions have greatly improved.Concrete points:1) It would be good to show some kind of error estimate for panel 1F (which might require normalizing to each replicate and not each treatment group), and this reviewer is not clear on how edgeR would be used to calculate a p value for this ratio, as is stated in the legend, or otherwise how what statistical test was used here?

We now included the SEM in the figure from n=3 independent samples for each condition, and further clarified how edgeR is applied to calculate the p values for the ratios in the legend as well as in the Materials and methods section: “Differential abundance analysis of cell clusters was done using the negative binomial generalized linear model (NB GLM) in EdgeR to calculate p-values. Adapting the differential abundance analysis used for mass cytometry data, we inputted counts as cells per sample instead of counts being reads per gene, and used the “glmQLFTest” function in EdgeR (Lun et al., 2016, Robinson et al., 2010).”

2) It would be useful for interpretability of Figure 6A to see the absolute # of MAC2 pos cells in the homeostatic vs repopulating clusters (or to refer to the panel where this was calculated, in the case that I missed this).

We have now included the absolute number of MAC2 positive cells in the homeostatic (672) vs. repopulating (187+1545) microglial populations.

Here are the numbers:

Cluster 1 and 2: 672,

Cluster 3 and 4: 187,

Cluster 5: 1545.

Subjective points:1) Language around Csf1r independence: The authors very effectively refined language around whether residual microglia are "Csf1r independent" or not. There is one sentence in the Abstract that still states that "a small subset of microglia in mouse brains can survive without Csf1r signaling", this has not been proven in prior studies, including the Pollard study in Csf1r KO mice, because this study did not test whether the IBA1 pos cells observed actually "survived" or were transient (for example, monocytes engrafting the brain and undergoing apoptosis).

We agree that it is likely some of the IBA1 positive cells do not survive and are “transient”. As we reported in our previous study, elevated apoptosis was observed during repopulation to “prune” the excess number of microglia (Zhan et al., 2019). Nevertheless, the large number of microglia at the end of repopulation indicates the number of surviving cells far outweighs those that do not survive (undergo apoptosis). Therefore, we think it is reasonable to state that "a small subset of microglia in mouse brains can survive without *Csf1r* signaling”.

2) "Progenitor signatures": The authors use the word "progenitor" to describe the young/embryonic microglial signatures used in comparisons in Figure 6 and supplement from the Matcovitch-Natan study, and describe MAC2+ cells as having "progenitor signatures." Strictly speaking, it is my impression that "progenitors" only refers to the "YS" population, not the E1 population and thereafter (which are no longer progenitors, but instead "early embryonic microglia"). They also claim that "progenitor-like" cells are present in the homeostatic brain. I would suggest changing the language to reflect the fact that microglia are not thought to have progenitors, and rather to self renew, acting as their own progenitor. Again, this is minor, I raise it only with goal of increasing precision of language since the subject of microglial progenitors is contentious, and in lumping together true yolk sac progenitors with early microglia, this may cloud the debate.

We thank the reviewer for helping us to be more precise in our description. We agree that YS population is strictly progenitor population, and have changed “progenitor signature” to “immature signature” in Figure 6 and Figure 7. However, since the upregulated DEGs of MAC2+ cells in the homeostatic brain exhibit substantial YS signatures (>20%), we believe that the use of “progenitor-like” is appropriate in some places since this term includes both signatures of YS and those of early embryonic development.

[Editors' note: further revisions were suggested prior to acceptance, as described below.]

The authors have addressed nearly all stated concerns raised by prior reviewers and significantly improved this manuscript. What remains unaddressed is definitive verification that cluster 5 cells (where the great majority of Mac2+ cells reside) are parenchymal microglia as opposed to border macrophages or something else. I think this remains a substantive concern as the crux of this paper as written is that Mac2+ microglia are resistant to CSF1R inhibition, and this conclusion seems to depend on whether cluster 5 cells are microglia. Some of these cells do express Cx3cr1 and Iba, and many CD74, but these markers are unfortunately not specific to microglia.

We thank the reviewer for the positive comments about the second revision. The reviewers’ concern prompted us to illustrate microglial signatures of Mac2+ subclusters more explicitly. After removing the outlier sample (D2-3) and performing re-clustering, Mac2+ cells are mainly located in Cluster-1 and Cluster-4. In both Cluster-1 and Cluster-4, expression of specific microglial markers, such as *Tmem119* and *P2y12*, which are also markers for homeostatic microglia, is comparable to other subclusters in the control condition (Revised Figure 3—figure supplement 1), confirming that both subclusters are indeed microglia. The levels of these homeostatic markers were downregulated in D0 and D2 conditions, which is one of the major findings of this manuscript (Figure 3).

Revisions expected in follow-up work:1) What remains unaddressed is definitive verification that cluster 5 cells (where the great majority of Mac2+ cells reside) are parenchymal microglia as opposed to border macrophages or something else. This is a a key point to many interpretations and conclusions is that the cells in cluster 5 are indeed microglia, since they are the predominant source of mac2/lgals3 pos cells. An alternative to showing this would be to assume they are not microglia and exclude them from analyses of repopulation microglia, but the authors did not take this route. Additionally, there is no significant difference in the sequencing data between Ctrl, D0, and D2 in abundance of Cluster 5 (Figure 1F). In fact based on that figure it appears that one mouse is a significant outlier for that cluster. Please address.

We thank the reviewer for pointing out the possibility that sample D2-3 could be an outlier. As shown with the UMAP and the cell distribution bar graph (Author response image 2), we agreed that the D2-3 sample is indeed an outlier, and have thus taken out this sample and performed re-clustering.

**Author response image 2. sa2fig2:** (a) UMAP split by samples showing that D2-3 had fewer cells than the other samples. (b) Ratio of cells from three treatment groups distributed in each cluster. Ratio of cells was calculated by normalizing to the total number of cells captured in each treatment group. Red dots represent sample D2-3.

Although there is no significant difference in the abundance of the main cluster with Mac2+ cells (Cluster-4 after re-clustering), the expression of *Lgals3* is significantly upregulated in D0 and D2 compared to Ctrl (revised Figure 4C).

2) When the authors remove clusters of contaminating cells from the analysis, it appears that they did not re-cluster the data. The UMAP plots are the same but without the contaminating clusters. This is not a valid analysis as the original clustering is based on variable genes and variable PCs from data that included all of the cells. Please address.

We thank the reviewer for this critical comment. We have performed re-clustering after removing the clusters of contaminating (non-microglial) cells and found that the new UMAP shows a promising separation between the homeostatic cluster and all others.

Reviewer #4:The authors have addressed nearly all stated concerns raised by prior reviewers and significantly improved this manuscript. What remains unaddressed is definitive verification that cluster 5 cells (where the great majority of Mac2+ cells reside) are parenchymal microglia as opposed to border macrophages or something else. I think this remains a substantive concern as the crux of this paper as written is that Mac2+ microglia are resistant to CSF1R inhibition, and this conclusion seems to depend on whether cluster 5 cells are microglia. Some of these cells do express Cx3cr1 and Iba, and many CD74, but these markers are unfortunately not specific to microglia.

We thank the reviewer for the supportive comment. As described in the response to reviewer 1, after removing the outlier sample (D2-3) and performing re-clustering, Mac2+ cells are mainly located in Cluster-1 and Cluster-4. In both Cluster-1 and Cluster-4, expression of specific microglial markers, such as *Tmem119* and *P2y12*, which are also markers for homeostatic microglia, is comparable to other subclusters in the control condition, confirming that both subclusters are indeed microglia (Revised Figure 3—figure supplement 1).

**References:**

Lun, A.T.L., Richard, A.C., and Marioni, J.C. (2017). Testing for differential abundance in mass cytometry data. Nat Methods *14*, 707-709.